# EvA: Evolutionary Attacks on Graphs

**Sadegh Akhondzadeh**[*1]    **Soroush H. Zargarbashi**[*2]    **Jimin Cao**[1]    **Aleksandar Bojchevski**[1]
[1] University of Cologne, [2] CISPA Helmholtz Center for Information Security
[akhondzadeh, zargarbashi, jcao, bojchevski]@cs.uni-koeln.de

## ABSTRACT

Even a slight perturbation in the graph structure can cause a significant drop in the accuracy of graph neural networks (GNNs). Most existing attacks leverage gradient information to perturb edges. This relaxes the attack's optimization problem from a discrete to a continuous space, resulting in solutions far from optimal. It also prevents the adaptability of the attack to non-differentiable objectives. Instead, we introduce a few simple, yet effective, enhancements of an evolutionary-based algorithm to solve the discrete optimization problem directly. Our Evolutionary Attack (EvA) works with any black-box model and objective, eliminating the need for a differentiable proxy loss. This allows us to design two novel attacks that reduce the effectiveness of robustness certificates and break conformal sets. EvA uses sparse representations to significantly reduce memory requirements and scale to larger graphs. We also introduce a divide and conquer strategy that improves both EvA and existing gradient-based attacks. Among our experiments, EvA shows ∼11% additional drop in accuracy on average compared to the best previous attack, revealing significant untapped potential in designing attacks.

## 1    INTRODUCTION

Given the widespread applications of graph neural networks (GNNs), it's crucial to study their robustness to natural and adversarial noise. In node classification, GNNs leverage the edge information to improve their performance. However, adding or removing a few edges can drastically decrease their accuracy, even below the performance of an MLP that ignores the graph structure entirely. The vast majority of adversarial attacks on the graph structure are gradient-based. However, gradient-based attacks face several challenges in this setting: (i) To tackle the original discrete combinatorial optimization problem we have to relax the domain from $\{0, 1\}$ to $[0, 1]$; (ii) The gradients only provide local information and cannot accurately reflect the actual loss landscape when edges are flipped (see Fig. 1 [Left]); (iii) Similarly, the gradient only reflects the effect of flipping *a single* edge at a time, but the effect on the loss can be different (even opposite) when two or more edges are flipped simultaneously (see Fig. 1 [Middle]); (iv) We need a differentiable proxy loss function since the original attack objective is often not differentiable (e.g. accuracy). A common choice is cross-entropy which is suboptimal as a proxy (Geisler et al., 2023); (v) White-box access to the model is necessary, which limits the applicability or requires surrogate models; (vi) Defense against such attacks might carry a false sense of security by only obfuscating gradients (Athalye et al., 2018; Geisler et al., 2023); (vii) Although the adjacency matrix is often sparse, the gradients w.r.t. it are not. Therefore, the memory complexity of these attacks grows quadratically w.r.t the number of nodes, for which tricks like block coordinate descent are needed (Geisler et al., 2021).

These challenges suggest that we should try to directly solve the original (combinatorial discrete) optimization problem, and not to rely on differentiation. A natural alternative is search. Indeed, Dai et al. (2018a) implemented a baseline genetic-based search for attacking the edges. However, their approach was not competitive with gradient-based attacks, largely due to poor design choices in the loss function and mutation strategies. While search-based attacks have been promptly forgotten since, we show that by carefully designing the components of a meta-heuristic pipeline we can outperform state-of-the-art gradient-based attacks by a significant margin. As shown in Fig. 1[Right], EvA not only outperforms the previous method based on a genetic algorithm, but also outperforms PRBCD, the previous state-of-the-art, by a large margin. Our model-agnostic evolutionary attack (EvA) explores

---

*Equal Contributions. Our implementation is available at the github.com/UoC-tail/EvA

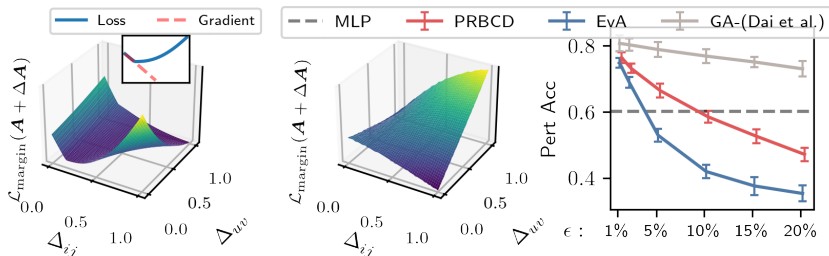

Figure 1: We compute $\mathcal{L}_{\mathrm{margin}}(\boldsymbol{A} + \Delta\boldsymbol{A})$ where $\Delta\boldsymbol{A} = \boldsymbol{e}_i\boldsymbol{e}_j^\top\Delta_{ij} + \boldsymbol{e}_u\boldsymbol{e}_v^\top\Delta_{uv}$ and $\boldsymbol{e}_i$ is the $i$-the cannonical vector. [Left] The loss landscape is non-linear, and the gradient does not always indicate the loss direction when we flip an edge (e.g. gradient suggests decrease, but loss increases). [Middle] Due to non-convexity, the effect of flipping each edge separately (e.g. loss decreases) can differ from flipping both edges simultaneously (e.g. loss increases). This happens for many edges (§ B). [Right] EvA does not suffer from this issue and outperforms both PRBCD and search-based GA attacks.

the space of possible perturbations with a genetic algorithm (GA) without gradient information. We avoid domain relaxation by directly minimizing the (non-differentiable) accuracy over the space of binary matrices. Our attack easily extends to other objectives. We show this by defining two novel attacks on graphs that break conformal guarantees or reduce the effectiveness of robustness certificates (see § 4). Importantly, this extension is automatic, we only need black-box access to the objective, making our attack adaptive (Mujkanovic et al., 2023). In contrast, gradient-based attacks for these new objectives require substantial additional effort (e.g. to tailor the right relaxations).

EvA requires $\mathcal{O}(\epsilon \cdot E \cdot P)$ memory where $\epsilon$ is the perturbation budget, $E$ is the number of edges and $P$ is the population size. Since $P$ is a small constant, we can simplify to $\mathcal{O}(\epsilon \cdot E)$. Given more time or more memory we can increase our performance due to the open-ended nature of the search, unlike PRBCD Geisler et al. (2021). For larger graphs where the search space is considerably larger, we apply a divide and conquer strategy that improves both PRBCD and EvA, with EvA still outperforming.

To summarize our contributions: (i) We carefully design the components of the GA, including a targeted adaptive mutation strategy and a better encoding, which leads to serious improvements ($\sim$11% additional drop in accuracy on average compared to the SOTA attack and up to $\sim$40% additional drop compared to the baseline GA attack); (ii) We broaden the scope of graph adversarial evaluation by attacking post-hoc guarantees such as conformal prediction and robustness certificates; (iii) To scale to larger graphs, we introduce a divide and conquer strategy that benefits both EvA and gradient-based attacks; (iv) We extend our attack to support local (per node) constraints with an efficient local projection. Our results caution against over-reliance on gradient-based attacks and show that search-based strategies remain a powerful and practical attack paradigm.

## 2 BACKGROUND AND RELATED WORK

**Problem setup.** We focus on attacking the semi-supervised node classification task via perturbing a small number of edges. We are given a graph $\mathcal{G} = (\boldsymbol{X}, \boldsymbol{A}, \boldsymbol{y})$ where $\boldsymbol{X}$ is the features matrix assigning a feature vector $\boldsymbol{x}_i$ to each node $v_i$ in the graph, $\boldsymbol{A}$ is the adjacency matrix (often sparse) representing the set of edges $\mathcal{E}$, and $\boldsymbol{y}$ is the partially observable vector of labels. Nodes are partitioned into labeled and unlabeled sets $\mathcal{V} = \mathcal{V}_l \cup \mathcal{V}_u$. The GNN is trained on a clean initial subgraph $\mathcal{G}_{\mathrm{tr}}$ that includes the labeled nodes. Following Gosch et al. (2024) we avoid the transductive setup ($\mathcal{G}_{\mathrm{tr}} = \mathcal{G}$) since perfect robustness can be achieved there by only memorizing the clean graph during training. They show that adversarial and self training also show a false sense of robustness in that setup for the same reason. Instead, we focus on inductive learning where a model $f$ is trained on an induced subgraph $\mathcal{G}_{\mathrm{tr}} \subseteq \mathcal{G}$, validated on $\mathcal{G}_{\mathrm{val}} \subseteq \mathcal{G}$ and tested on $\mathcal{G}_{\mathrm{test}}$ where $\mathcal{G}_{\mathrm{tr}} \subset \mathcal{G}_{\mathrm{val}} \subset \mathcal{G}_{\mathrm{test}} = \mathcal{G}$.

**Threat model.** We optimize over a perturbation matrix $\boldsymbol{P} \in \{0,1\}^{n \times n}$ that flips entries of the adjacency matrix $\tilde{\boldsymbol{A}} = \boldsymbol{A} \oplus \boldsymbol{P}$, where $n = |\mathcal{V}|$, and $\oplus$ is the element-wise XOR operator. For a given function $f$ as the GNN model, and any generic loss function $\mathcal{L}$, the objective is

$$\boldsymbol{P} = \underset{\boldsymbol{P}}{\arg\max} \quad \mathcal{L}(f(\mathcal{G}(\boldsymbol{X}, \boldsymbol{A} \oplus \boldsymbol{P}))_{\mathrm{att}}, \boldsymbol{y}_{\mathrm{att}}) \qquad s.t. \quad \mathbf{1}_N\boldsymbol{P}\mathbf{1}_N^\top \leq \epsilon \cdot |\mathcal{E}[\mathcal{V}_{\mathrm{att}} : \mathcal{V}]| \qquad (1)$$

Here $f(\cdot)_{\text{att}}$ is the vector of predictions for the subset of nodes $\mathcal{V}_{\text{att}}$ that are under attack. In "targeted" attacks $\mathcal{V}_{\text{att}}$ is a singleton. $\mathcal{L}$ is negative accuracy or any other objective function explained in § 4. To keep the perturbations imperceptible, we assume that the adversary can only perturb up to $\delta := \epsilon \cdot |\mathcal{E}[\mathcal{V}_{\text{att}} : \mathcal{V}]|$ edges where $\mathcal{E}[\mathcal{A} : \mathcal{B}]$ is the subset of edges between nodes in $\mathcal{A}$ and $\mathcal{B}$. Eq. 1 can include more constraints like the local constraint from Gosch et al. (2023) restricting perturbations not to increase node degrees by more than a fraction (e.g $e_{\text{loc}} = 0.5$) of their original value.

**Related Work.** We study evasion attacks with both global and targeted objectives, where perturbations are introduced only at test time. The goal is either to reduce the model's overall accuracy or to induce the misclassification of a specific node. Among gradient-based methods, PRBCD (Geisler et al., 2021) and LRBCD (Gosch et al., 2024) represent the current state of the art. Both attacks compute gradients of the $\tanh$-margin loss with respect to the adjacency matrix, employing block-coordinate descent. Perturbation edges are then sampled based on these gradients. To handle local degree constraints, LRBCD incorporates a local projection step, greedily selecting edges (in descending order of gradient score) while ensuring that the constraints are not violated.

Beyond gradient-based methods, alternative approaches have also been explored. For example, Dai et al. (2018a) proposed a simple evolutionary attack as a baseline for their reinforcement learning-based method. However, these early strategies have since been surpassed by gradient-based techniques. Building on this progress, we redesign key components of the search process, achieving significant improvements over prior evolutionary attacks. Moreover, our method, EvA, scales effectively to large graphs and naturally extends to novel attack objectives. Other heuristic-based attacks, relying on node degree, centrality, or related metrics (Zhang et al., 2024; 2023; Wang et al., 2023), have also been proposed, but they similarly fail to outperform the current state-of-the-art methods. A more detailed discussion of related work can be found in § A.

## 3 EvA: Evolutionary Attack

**Components of EvA.** As shown in Fig. 1 gradients can be misleading which motivates us to explore search-based attacks. We employ a genetic algorithm (GA) (Holland, 1984) that starts with an initial population of candidate solutions that we iteratively refine. The improvement is driven by the fitness function, and the crossover and mutation operators. Here we provide a brief overview. For the detailed technical description see § D. Each individual in the population specifies a set of edges to be flipped. We implement an efficient $\mathcal{O}(1)$ mapping from a 1D index to 2D edges, while ensuring undirected flips. The *fitness* function that evaluates each individual should correlate with the objective in Eq. 1 and provide sufficient sensitivity across candidates. For global attacks, we use the model's accuracy on the perturbed graph. In § 4 we explore better alternatives for local and targeted attacks. We generate new candidates from two individuals via a *crossover* operator, which concatenates parent segments. Parents are chosen through tournament selection ($n_{\text{tour}}$ random samples from which the two fittest are retained). The baseline *mutation* operator randomly replaces each index with some probability. We design significantly better mutations below.

**Sparse encoding of the attack.** A simple way to represent a perturbation is a boolean vector of size $N^2$ encoding which edges are flipped. It costs $\mathcal{O}(|\mathcal{S}|N^2)$ space from memory where $\mathcal{S}$ is the population. This representation is not aware of the sparsity in $\boldsymbol{A}$. Instead we represent each candidate as a list of indices to be toggled in the adjacency matrix, we store sparse representation of $\boldsymbol{P}$. With this we account for the sparsity and reduce the complexity to $\mathcal{O}(|\mathcal{S}| \cdot \delta)$ where $\delta = \lfloor \epsilon \cdot |\mathcal{E}[\mathcal{V}_{\text{att}} : \mathcal{V}]| \rfloor$ – candidates in the population $\boldsymbol{z} \in \mathcal{S}$ are vectors of $\delta$ dimensions with each entity as an index in adjacency matrix $\boldsymbol{z}[i] \in \{1, \cdots, n(n-1)/2\}$ with $n = |\mathcal{V}|$. Our mapping $\Pi$ (as discussed in § D) is a diagonal enumeration of an upper triangular $n \times n$ matrix. For simplicity, we let the perturbation vector to contain repeated elements. During the evaluation of the vector, we transform it to a perturbation matrix $\boldsymbol{P_z}$, with which we compute $\tilde{\boldsymbol{A}} = \boldsymbol{A} \oplus \boldsymbol{P_z}$. We compute all steps with sparse representation, where each candidate takes $\mathcal{O}(\delta)$ space. Moreover, with this encoding, we directly enforce the global budget since the size of each individual in the population is at most the number of allowed perturbations by design.

**Accuracy vs alternative surrogate losses.** To understand the effect of the loss on the attack, we conducted an ablation study to compare accuracy and common surrogate objectives (cross-entropy, and margin-based loss) as the fitness function in EvA. As in Fig. 2 (left), cross-entropy does not use the attack budget effectively, while margin-based loss shows to be well-correlated. Intuitively, since the goal is to misclassify as many nodes as possible, the aggregated cross-entropy loss can

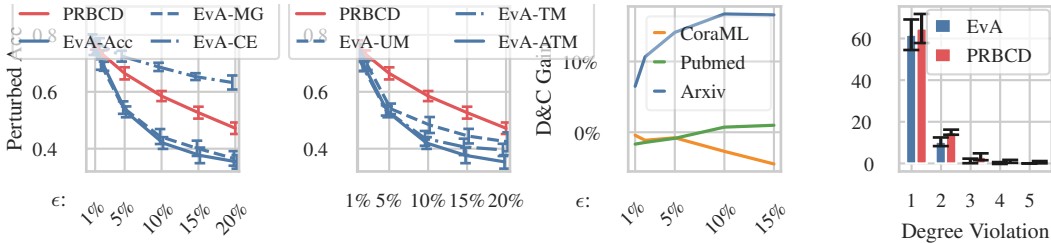

Figure 2: [Left] EvA's performance with different objective functions and [Middle left] mutation strategies. [Middle right] Effect of D&C on the performance of EvA for different datasets. [Right] The number of violations from local constraints by EvA, and PRBCD ($\epsilon_{\text{loc}} = 0.5$).

waste perturbation budget by overly focusing on already misclassified nodes, rather than maximizing new misclassifications. This effect was studied in depth by Geisler et al. (2023), which motivated them to introduce the Tanh-Margin loss which mitigates this issue. Still among all fitness functions, accuracy itself performs better. Since PRBCD uses the Tanh-Margin loss, the large gap between EvA, and PRBCD suggests that the quality of loss is not the only reason behind EvA's effectiveness. We hypothesise that EvA, leveraging the exploratory capabilities of GA, can explore the search space more effectively and avoid local optima, while PRBCD gets stuck.

**Drawbacks.** The mentioned setup is the baseline variant of EvA. Combined with cross-entropy as the fitness it is similar to a parallel and efficient implementation for Dai et al. (2018a) which is by far less effective (see Fig. 1). While the baseline (with accuracy) already outperforms SOTA Fig. 2, we enhance the search by introducing a better initial population and a mutation function that discards edges outside of the target's receptive field.

**Enhancing the search.** To enhance EvA, the key insight is that by restricting the search space to the receptive field of $\mathcal{V}_{\text{att}}$ (instead of the entire $\frac{n}{2}(n-1)$ edges), we eliminate less effective (or ineffective) perturbations from the search space. Perturbations that have both endpoints in the training subgraph can be easily reverted by memorization. Additionally, flipping edges outside of the receptive field of $\mathcal{V}_{\text{att}}$ is a waste of budget since they do not affect the prediction of $\mathcal{V}_{\text{att}}$. Similarly, we restrict the initial population to have at least one endpoint in $\mathcal{V}_{\text{att}}$. This is easily done by randomly sampling both endpoints, one inside $\mathcal{V}_{\text{att}}$ and one in $\mathcal{V}$, then mapping the edges back to the indices via $\Pi$. For larger graphs, as the search space increases quadratically to the number of nodes, we can apply a divide and conquer strategy by splitting $\mathcal{V}_{\text{att}}$ and running EvA on each chunk.

**Targeted and adaptive mutation.** Mutation is applied by selecting a set of perturbation indices (uniformly at random with probability $p$) from the population and changing them to another index. A naive implementation (the underline{u}niform underline{m}utation (UM)), adds random indices from anywhere in the entire graph. Similar to the initialization, we define the "underline{t}argeted underline{m}utation (TM)" by restricting the new mutated edge to have at least one end-point in $\mathcal{V}_{\text{att}}$. Furthermore, when the attack succeeds in altering a node's label, perturbing its connections does not increase the performance anymore. Hence, we exclude the already flipped nodes from the endpoint that was restricted to $\mathcal{V}_{\text{att}}$. Importantly, we still allow those nodes to connect with other nodes in $\mathcal{V}_{\text{att}}$ as they can contribute to the misclassification risk of other nodes. We refer to this approach as "underline{a}daptive underline{t}argeted underline{m}utation" (ATM). Remarkably, as shown in Fig. 2 (right), these modifications improve the effectiveness of EvA by a noticeable margin.

**Stacking perturbations.** EvA requires a forward pass per each candidate (each candidate of population). While maintaining the sparse representation of the graphs during all steps, we can use the remaining memory to combine $k$ candidates in form of a large graph of $k$ parts and evaluate all $k$ in a single forward pass. In practice, we can easily fit the entire population in one forward pass per iteration as one large graph.

**Effect of scaling.** The population size has a considerable impact on the performance of EvA by introducing diversity among the solutions, thus increasing exploration. To observe this effect, we do an ablation study on the population size and the number of iterations. For a fair comparison, we scale PRBCD separately by increasing the number of steps and the size of the block coordinate subspace. We exponentially increased the block size, starting from 0.5M up to 4 million Fig. 3[Left]. As

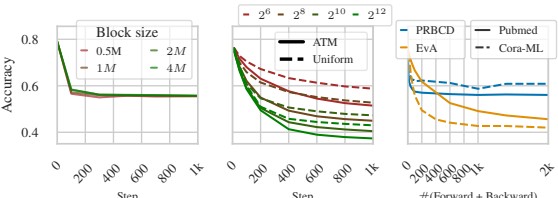
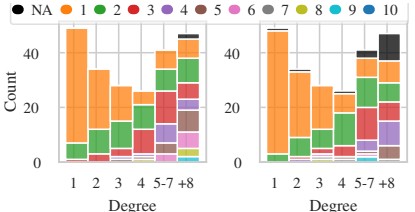

Figure 3: (From left to right) Scaling performance of PRBCD over various memory and iteration budgets, the effect of mutation on different resources on `Pubmed` and number of forward vs performance all with $\epsilon = 0.1\%$.

Figure 4: Number of perturbations (different colors) used by EvA [left] and PRBCD [right] to attack nodes of specific degrees. Black (NA) shows failed attacks.

shown in Fig. 3 [middle], increasing the population size, and then increasing the number of iterations improves EvA. In contrast, PRBCD does not achieve noticeable improvement by increasing the block size or the number of training steps (Fig. 3 (left,right)). This means that EvA leverages additional computational resources (either time or memory) while PRBCD does not show a considerable use of it. As a supplement to Fig. 2, in Fig. 3 [Middle] we show that using a better mutation function (here adaptive targeted mutation) consistently enhances performance across all population sizes and outperforms the uniform approach. Finally, in Fig. 3[Right], we compare the number of forward passes used in EvA with the number of forward and backward passes applied in PRBCD, and show their performance under approximately equal memory usage. Initially, PRBCD converges faster and outperforms EvA, but at larger scales EvA significantly surpasses PRBCD. We also compare EvA and SOTA for wall-clock time and memory in § D.3 showing similar results. In general, we see that EvA is more often Pareto optimal, and a broader range of time-memory-performance trade-offs.

**Divide and Conquer.** PRBCD uses the coordinate gradient descent to scale efficiently for larger graphs. Similarly we introduce a divide-and-conquer (D&C) approach to EvA. Here instead of attacking the entire $\mathcal{V}_{att}$ at once, we divide it into smaller subsets and sequentially attack each subset with a budget relative to the portion of the edges connected to it. After attacking a subset, we treat the modified graph as a starting point for the next one. The result for the final subset includes perturbations in all previous steps. At the end we re-evaluate the final graph with all perturbations combined. Our divide and conquer approach relies on a relaxation. For a budget of $\delta = \delta_1 + \delta_2$ over a set $\mathcal{V}_{att} = \mathcal{V}_1 \cup \mathcal{V}_2$, the standard attack searches in the space of $\binom{n}{2}^{\delta}$ possible perturbations aiming to decrease the accuracy over $\mathcal{V}_{att}$. However, with the divide and conquer approach, the attack searches among $\binom{n}{2}^{\delta_1} + \binom{n}{2}^{\delta_2}$ possible perturbations each aiming to attack $\mathcal{V}_1$, and $\mathcal{V}_2$ separately - first searching for optimal attack with a budget $\delta_1$ on $\mathcal{V}_1$ and then with $\delta_2$ on $\mathcal{V}_2$ given the attack applied on $\mathcal{V}_1$. Therefore, D&C explores an exponentially smaller subset of the search space. This calculation is for the uniform mutation, employing targeted mutation narrows the search space explored by the algorithm further. Since it also reduce the choices from $\binom{n}{2}^{\delta}$ to $\left(\frac{|\mathcal{V}_{att}|(2n-|\mathcal{V}_{att}|-1)}{2}\right)^{\delta}$. In practice we divide $\mathcal{V}_{att}$ to $k_{dc}$ subsets (see hyper-parameters in § E.4). Applying the D&C approach poses a trade-off: in smaller spaces EvA finds better solutions, while the relaxation in D&C can lead to solutions further from optimum. As shown in Fig. 2 (right) when the size of the graph and the budget $\delta$ grow, adding D&C to EvA helps substantially. As in Fig. 2 (right) it improves the result for large `Ogbn-Arxiv` by at least $\sim 8\%$ while the same approach is ineffective for smaller `CoraML` dataset. We further show that on large graphs D&C similarly helps PRBCD. Indeed, applying D&C for PRBCD helps to increase the performance while maintaining the same block-size (not increasing the required memory; see § D.4). A comparable block for 1-step PRBCD exceeds the memory limit. It is noteworthy that the randomized block-coordinate computation of gradients is also a relaxation.

## 4 LOCAL AND TARGETED ATTACKS & ATTACKING OTHER OBJECTIVES

**Local attacks.** Gosch et al. (2024) extend PRBCD to support additional "local" constraints where a perturbation is not allowed to increase the degree of a node more than a fraction of its original value. We need local constraints to enforce imperceptibility of the attack. For example, a perturbation might increase the degree of a node more than twice its original value while staying within the global

budget. Therefore, even within the global budget the structure of the graph (and therefore graph's structural semantics) can change drastically. They introduce the LRBCD attack which adds a local projection to PRBCD. In short, they sort edges in a decreasing order of probability (gradients), and iteratively add perturbations while the local constraint for both end-points of the modified edge is not violated. This continues until the global budget is exhausted. Even without enforcing this restriction, as shown in Fig. 2 (right) EvA introduces fewer degree violations compared to PRBCD, meaning that the perturbations added by EvA are more spread-out in the graph. We apply a local projection to EvA similar to LRBCD. Here, instead of using gradients for ranking, we use the frequency of the edge within the current population. We define $s(e) = \sum_{s \in \mathcal{S}} \mathbb{I}[e \in s]/|\mathcal{S}| + u$ as the frequency score where $u$ is a small uniform random value in $[0, 0.05]$. Here $u$ is added to break ties and introduce additional randomness, and $\mathcal{S}$ is the population at the current iteration. Our insight is that if an edge appears frequently within the population, it is likely to be useful for an attack, increasing the chance of candidates containing it to be selected as elite. After our local projection all constraints are guaranteed to be satisfied. For more diversity at initial iterations, we apply a random projection removing edges with a probability proportional to total degree violations on both sides. We apply this random removal for $t_{\text{warm}}$ iterations (a hyper-parameter discussed in § E.4).

**Node-targeted attacks.** Here the objective is to misclassify a specific node with as minimal change to the structure as possible. Using EvA with the global setup does not work in this case. On a single node the accuracy has only two values $0$ or $1$ – small changes in the solution do not result in (even minor) changes in the fitness score. With the 0-1 accuracy objective, random search and GA are practically equivalent as there is no indication of what combination of edges are closer to breaking the prediction of a particular node – all non-successful combinations are equally evaluated with $1$. Instead we use the proxy $\tanh$-margin loss as the fitness function. This loss function changes as we perturb the receptive field of the targeted node. Note, for general (non-targeted) attacks the $\tanh$-margin loss improves performance over the cross-entropy loss, however, using accuracy (for larger $|\mathcal{V}_{\text{att}}|$) as fitness is slightly better as shown on Fig. 2 (left). Fig. 4 compares EvA, and the state-of-the art attack PRBCD on targeted attacks.

**Other objectives.** For non-differentiable objectives (e.g. accuracy), gradient-based attacks need a differentiable surrogate approximating it. As discussed in § 3, for accuracy (common setup) several works proposed various surrogates. This is similarly challenging to propose gradient-based attacks for novel objectives that are complicated and include several non-differentiable components (e.g., quantile computation or majority voting from Monte Carlo samples). Since our method nullifies the need for information from gradients, we can easily optimize for novel complex objectives as long as they are sensitive to small changes in the search space. We define three new attacks on graphs: reducing the certified ratio of a smoothing-based model, decreasing coverage, and increasing the set size of conformal sets. A detailed explanation of randomized smoothing-based certificates and conformal prediction, which underpin the certified ratio objective and conformal prediction, is provided in § A.1 and § A.2 respectively.

**Attacking smoothing-based certificate.** Assuming the certified ratio is a notion of a trustworthy prediction, one possible adversarial objective is to reduce the number of nodes that are certified (a.k.a. certified ratio) while maintaining the same clean accuracy. While the operations include non-differentiable steps we can directly set the certified ratio (fraction of nodes that are certified within a determined threat model) as the objective of EvA. Whether a node is certified reduces to whether the smooth classifier returns a probability above $\overline{p}$ where $\min_{\tilde{x} \in \mathcal{B}(x)} g(x) \geq 0.5$ constrained to $g(x) = \overline{p}$. Many smoothing-based certificates are computed at canonical points (they are only a function of probability not the input) and they are non-decreasing to $\overline{p}$. Hence, we find $\overline{p}$ via binary search. Thus, our objective is to decrease the number of vertices with smooth probability above $\overline{p}$. A naive implementation of EvA for this objective is to compute the certified ratio given new MC samples for each candidate. This increases the runtime of our algorithm by a factor of $n_{\text{MC}}$, as each perturbation requires $n_{\text{MC}}$ forward passes. Inside the attack, statistical rigor is not crucial. Therefore, we employ an efficient sampling strategy where we start with initial samples from clean $\boldsymbol{A}$, and for each perturbation, we only resample for the edges in $\tilde{\boldsymbol{A}} \oplus \boldsymbol{A}$. We use the stacked inference technique (see § 3) on MC samples which ultimately reduces the computation to one inference per each perturbation $\tilde{\boldsymbol{A}}$.

**Attacking conformal prediction.** A common threat model for CP is to decrease the empirical coverage (far from the guarantee) by perturbing the test input. We propose a similar attack where the

adversary changes the edge structure of the graph in order to decrease the coverage. This process is again not directly differentiable (for steps like computing the quantile and comparison of values) which is not a problem for EvA. In our experimental setup, the defender calibrates on a random subset of $\mathcal{V}_u$ (besides the test, this is the only set with labels unseen by the model). Assuming that the unlabeled and test nodes are originally exchangeable (node-exchangeability), the conformal guarantee is valid in the inductive setup upon recalibration on the clean graph. By perturbing the edge structure we can easily break this guarantee. Therefore our objective is to change the edge structure such that the coverage is minimized. Intuitively, this requires maximizing the distribution shift between the test and calibration scores. As we know that the calibration set is an exchangeable (random) subset of $\mathcal{V}_u$, we set the entire $\mathcal{V}_u$ as the calibration set during the attack. Due to exchangeability we expect a similar effect from our perturbation for any random subset as well (Berti and Rigo, 1997). Finally, the objective is to decrease the coverage over $\mathcal{V}_{att}$ given $\mathcal{V}_u$ as the calibration set. To the best of our knowledge, so far this is the only adversarial attack on the graph structure to break conformal inductive GNNs. Similarly, by changing the objective to the negative average set size, we can attack the usability of prediction sets (see Fig. 6).

## 5 EMPIRICAL RESULTS

With our empirical evaluations we show that current gradient-based attacks are still very far from optimal since EvA outperforms them by a notable margin. EvA inherently results in attacks with a smaller local change in each node's degree (even without posing local constraints)Fig. 2[right]. And further we can apply EvA to attacks with local constraints as well. With divide and conquer, EvA is able to scale to larger graphs (e.g. `Ogbn-Arxiv`) and outperform SOTA for those graphs as well. With the black-box nature of the attack we easily extend the score of EvA to novel objectives introducing the first attack to reduce the certified ratio or break conformal sets on graphs.

**Experimental setup.** We evaluate EvA on common graph datasets: `CoraML` (McCallum et al., 2004), `Citeseer` (Sen et al., 2008), and `Pubmed` (Namata et al., 2012). Shchur et al. (2018) show that GNN evaluation is sensitive to the initial train/val/test split. Therefore, we averaged our results for each dataset/model over five different data splits. In contrast with common GNN attacks, Gosch et al. (2024) shows that the transductive setup carries a false sense of robustness. In other words, trivially one can gain perfect robustness just by memorizing the clean graph which is available before the attack; models with robust and self-training also show how to exploit this flaw. Following them, we report our results in an inductive setting. We divide graph nodes into four subsets: training, validation, and testing, each with 10% of the nodes and we leave the remaining 60% as unlabeled data. Following Lingam et al. (2023), we sample the train, validation and test nodes in exchangeably since it provides a more realistic setup compared to commonly used methods, such as sampling for training and validation with the same count for each class (i.e. stratified sampling). For completeness, in § C we compare attacks in the transductive setup and various sampling approaches. In all cases again EvA shows a more effective attack. Further information about the model and hyperparameters are in § E.

Table 1: Performance of different attack methods under varying budgets on `CoraML` dataset.

| Attack | 0.01 | 0.02 | 0.05 | 0.10 | 0.15 |
|--------|------|------|------|------|------|
| DICE   | 80.93 | 80.93 | 80.78 | 80.57 | 80.07 |
| PGA    | 79.58 | 76.92 | 70.94 | 64.62 | 60.46 |
| PGD    | 78.22 | 75.37 | 67.18 | 59.14 | 53.09 |
| GRPCD  | 78.07 | 75.08 | 66.76 | 58.29 | 54.80 |
| PRBCD  | 76.44 | 73.17 | 66.48 | 58.51 | 52.67 |
| EvA    | **74.80** | **68.97** | **52.95** | **41.99** | **37.65** |

Table 2: Performance of different defense models under various attack strengths on `CoraML`.

| Defense | Attack | 0.01 | 0.02 | 0.05 | 0.10 |
|---------|--------|------|------|------|------|
| GCNSVD | EvA | **0.70** | **0.64** | **0.54** | **0.41** |
|        | PRBCD | 0.76 | 0.75 | 0.73 | 0.70 |
| GNNGuard | EvA | **0.71** | **0.67** | **0.55** | **0.45** |
|          | PRBCD | 0.74 | 0.72 | 0.70 | 0.66 |
| GNNJaccard | EvA | **0.76** | **0.74** | **0.64** | **0.57** |
|            | PRBCD | 0.76 | 0.74 | 0.70 | 0.65 |
| Robust-GCN | EvA | **0.75** | **0.70** | **0.59** | **0.52** |
|            | PRBCD | 0.77 | 0.73 | 0.68 | 0.63 |
| Soft-Median | EvA | **0.75** | **0.72** | **0.69** | **0.62** |
|             | PRBCD | 0.77 | 0.76 | 0.73 | 0.68 |

**Attacking vanilla and robust models.** As shown in Fig. 5 and extensively in § C, EvA outperforms the SOTA attack PRBCD by a significant margin consistently across various datasets and models (vanilla and robust). Interestingly, we show that on many vanilla and robust models, for a very small budget $\epsilon \sim 0.05$, EvA drops the accuracy below the level of the MLP model. This is a condition where the model leveraging the structure works worse than a model that completely ignores edges.

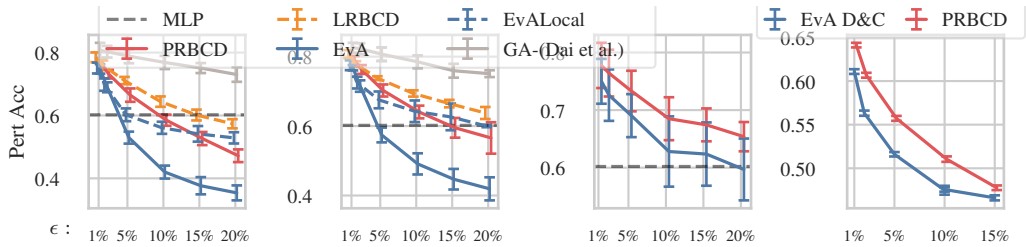

Figure 5: (Left to right) Performance on `CoraML` on Vanilla GCN, adversarially trained GCN using PRBCD, Soft-Median-GDC model. The right-most figure is GCN on `Ogbn-Arxiv`.

The SoftMedian model seems to show an inherent robustness to both EvA and PRBCD. Therefore, to break the model below the accuracy of MLP, we require $\geq 0.2$ perturbation budget. Even in the SoftMedian model, our attack is significantly more effective in comparison to PRBCD. Table 1 compares EvA with other attacks, showing that our attack outperforms all previous methods. We also provide additional results on different defense mechanisms in Table 2, which demonstrate that our attack can break them all. We also provide additional result with adversarial training. Table 14 (§ C) compares attacks against models with different adversarial training. We study the characteristics of the perturbed edges in § D.2.

**Scaling to larger graphs.** In Fig. 5, we also show that the EvA applied with our divide and conquer approach outperforms PRBCD for `Ogbn-Arxiv` dataset. Interestingly similar divide and conquer approach can significantly improve PRBCD as well; while still EvA is more effective. In § D.4, we compared PRBCD with block size 3M, 10M, alongside PRBCD and EvA with divide and conquer in a fair comparison. Notably PRBCD with the highest block size fitting in one GPU is still significantly less effective compared to any of the attacks combined with D&C.

**Additional datasets.** To show that EvA generalizes beyond citation graphs, in Table 3, we compare it with PRBCD on two co-purchase graphs, the AMAZON-PHOTO and AMAZON-COMPUTERS graph (Shchur et al., 2018). EvA is still better.

Table 3: Performance on non-citation graphs.

| Dataset | Attack | $\epsilon = 0.01$ | $\epsilon = 0.02$ | $\epsilon = 0.05$ | $\epsilon = 0.10$ |
|---|---|---|---|---|---|
| `photo` | PRBCD | $84.28 \pm 0.99$ | $81.12 \pm 1.45$ | $75.86 \pm 1.81$ | $71.58 \pm 1.80$ |
| | EvA | $\mathbf{80.43} \pm \mathbf{1.47}$ | $\mathbf{77.99} \pm \mathbf{2.04}$ | $\mathbf{72.73} \pm \mathbf{1.90}$ | $\mathbf{67.01} \pm \mathbf{1.76}$ |
| `computers` | PRBCD | $77.28 \pm 0.68$ | $73.56 \pm 0.55$ | $66.92 \pm 0.63$ | $61.99 \pm 0.59$ |
| | EvA | $\mathbf{72.94} \pm \mathbf{1.26}$ | $\mathbf{70.10} \pm \mathbf{1.85}$ | $\mathbf{65.70} \pm \mathbf{2.89}$ | $\mathbf{60.13} \pm \mathbf{3.72}$ |

**Local attacks.** Similarly, as shown in Fig. 5, and § C, EvA is consistently better than LRBCD. In § 4 discussed that we apply local projection as a mutation function. Interestingly as in Fig. 16 (right) even without local projection, EvA results in less violations of the local constraint.

**Targeted attack.** We perform targeted attacks on each node separately, with varying budgets from one to a maximum of 10 edges, until the prediction changes. We discussed in § 4, that here we used $\tanh$-Margin proxy loss since accuracy on one node is not sensitive to small changes. Fig. 4 compares EvA and PRBCD in targeted attack. Our results show that PRBCD performs better with a budget of one, but is outperformed by EvA for budgets of two and higher. For instance, on the `CoraML` dataset PRBCD fails to modify 16 nodes with a maximum of 10 changes (NA, black), whereas this number is reduced to only 2 nodes for EvA. This result is expected due to the combinatorial nature of the problem: for budgets up to two, a greedy approach can find the optimal solution, but as the budget increases beyond three, the problem becomes significantly more complex. This is also in line with our first motivation that the gradient ignores the interaction effect of flipping multiple edges simultaneously. Fig. 1 [middle] is an instance when the gradient direction individually has the same direction, but the loss when flipping both is in the opposite direction. This effect can become even more problematic when one flips more edges.

**Attacking novel objectives.** In Fig. 6 (mid-right and right) we performance of EvA with the objective to reduce the certified ratio. The plots are for certificate on $\mathbf{A}$ (mid-right) with $(p_+ = 0.001, p_- = 0.4)$, and $\mathbf{X}$ (right) with $(p_+ = 0.01, p_- = 0.6)$ with sparse smoothing (Bojchevski et al., 2020). Here $p_+$, and $p-$ are Bernoulli parameters of flipping a zero or one. In both plots we report the

Figure 6: (From left to right) conformal coverage, and conformal set size on vanilla and adversarially trained GCN. The certificate attack for certified ratio on $A$ and $X$ evaluated on GPRGNN adversarially trained using PRBCD. All plots are for `CoraML`.

result for $\mathcal{B}_{0,3}$ which means 0 additions and 3 deletions. While we aim to decrease the certified ratio, a direct outcome is that the certified accuracy drops. For a 5% budget, the certified accuracy drops below MLP. MLP is a baseline model with full robustness to edge perturbations (since it discards the adjacency information completely). While reducing the certified ratio, interestingly the smooth model's accuracy remains the same. Hence, evaluating the model on a holdout labeled set does not reveal that the input graph is attacked. We report the first structure attack on an inductive conformal GNN. As shown in Fig. 6 (right) the coverage drops quickly as we increase the perturbation budget. As expected, in an adversarially trained model, we observe a slower decrease in the empirical coverage. Alternatively in Fig. 6 (middle) we increase the average set size since showing that that both vanilla and robust models are vulnerable to this attack.

**Ablation study on the effect of our different GA extensions.** To emphasize the effects of our simple yet effective enhancements, we provide the following ablation studies. In Table 4, we show the effect of each enhancement individually and then together (EvA) on the `CoraML` dataset. Furthermore, in Table 5, we report the effect of our sparse encoding (SE) and D&C on the larger `Ogbn-Arxiv` dataset. As shown, all of our simple enhancements provide a significant effect (individually and jointly).

Table 4: The effect of our adaptive targeted mutation (ATM) and the fitness function on `CoraML`.

| $\epsilon$ | 0.01 | 0.02 | 0.05 | 0.10 | 0.15 |
|---|---|---|---|---|---|
| $(*)$ Dai et al. | 80.71 | 80.28 | 78.86 | 76.86 | 75.08 |
| $(*)$ + ATM | 78.50 | 76.65 | 72.52 | 68.75 | 65.33 |
| $(*)$ + $\mathcal{L}_{acc}$ | 75.08 | 69.39 | 54.02 | 48.32 | 44.41 |
| EvA (+ both) | 74.80 | 68.96 | 52.95 | 41.99 | 37.65 |

Table 5: Effect of our sparse encoding and D&C on the large `Ogbn-Arxiv` dataset.

| $\epsilon$ | 0.01 | 0.02 | 0.05 | 0.10 |
|---|---|---|---|---|
| Dai et al. | OOM | OOM | OOM | OOM |
| Dai et al. + SE | 69.79 | 69.56 | 68.81 | 67.77 |
| EvA | 66.86 | 66.80 | 65.18 | 63.51 |
| EvA + D&C | 61.08 | 56.31 | 51.60 | 47.56 |

## 6 CONCLUSION

We introduce EvA, an adversarial attack on the graph structure using a genetic algorithm. Unlike gradient-based methods, our black-box approach directly optimizes the adversary's objective (e.g. the model's accuracy). This flexibility allows for more complex adversarial goals – we demonstrate successful attacks that decrease certified robustness and degrade conformal prediction performance. To ensure scalability, we propose an efficient encoding that ties memory complexity to the perturbation budget and a divide-and-conquer strategy that improves performance on large graphs for both our method and baselines like PRBCD. We also show that due to the open-ended characteristic of the search, for more computational resources (time and memory) we can always improve our results. Given the significant decrease in the model's accuracy by applying EvA, we highlight that even SOTA gradient-based attacks are far from optimal. Our main message is that search-based attacks are underexplored yet powerful as shown by our results.

**Limitations.** We use an off-the-shelf genetic algorithm. Surely, there is room for designing search algorithms specific to the domain of the problem beyond our extensions, or even hybrids of gradient and evolutionary search. EvA uses many forward passes through the model which can be unrealistic in some attack scenarios. We leave the design of a further query-efficient variant for the future.

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

# A    Supplementary to Related Work

We focus on evasion attacks where perturbations are made after the model's training. Based on the domain, (evasion) attacks can be further be categorized into global (aiming to flip the prediction of a subset of nodes) and targeted attacks (aiming at a single node). Our attack applies on edge-structure similar to Xu et al. (2019); Zügner et al. (2018); Geisler et al. (2023; 2021); Gosch et al. (2024). Orthogonal to this scope, various other graph attacks are proposed in the literature including node-injection attacks (Ju et al., 2023), poisoning (Zügner et al., 2020; Lingam et al., 2023; Zügner et al., 2018), and attacking attributes (Zügner et al., 2018). Inspired by techniques used on continuous data, Xu et al. (2019); Zügner et al. (2018); Geisler et al. (2023) utilize gradients to approximate perturbations on inherently discrete edges. As the adjacency matrix can grow significantly larger than images, applying a PGD-like attack becomes challenging for larger graphs. To remedy that Geisler et al. (2021) proposes a block-coordinate computation of the derivatives, and Gosch et al. (2024) applies a greedy projection to apply local constraints.

Orthogonally, Dai et al. (2018b) use reinforcement learning to refine their attack and disrupt the learning process of GNNs Sun et al. (2023). They also introduce a genetic algorithm attack as a baseline; however, they did not design the components of GA carefully. In § 3 we design GA components (mutation, local projection, etc) which outperform recent gradient based attacks. Recently new attacks relying on heuristics such as node degree, centrality, etc have been proposed (e.g. Zhang et al. (2024; 2023); Wang et al. (2023)), however they don't outperform the SOTA.

**Gradient-based attacks.** A common class of attacks compute the gradient of the objective w.r.t. $\boldsymbol{A}$. This requires a relaxation on the domain of $\boldsymbol{A}$ from $\{0,1\}^{n \times n}$ to $[0,1]^{n \times n}$. For non-differentiable objectives like accuracy differentiable surrogates like the categorical cross entropy or $\tanh$-margin (Geisler et al., 2023) are used instead. The algorithm is to iteratively compute the gradients and update the perturbation matrix. Finally, based on the continuous perturbation matrix edges are either sampled or rounded to the binary domain.

**Black-Box attack.** The literature on black-box attacks on graphs remains relatively underexplored. Some existing works focus on poisoning attacks (Chang et al., 2020). Other studies, such as Waniek et al. (2018) and Xu et al. (2019), propose heuristic attacks based on the graph's topology, but their performance is significantly lower than that of white-box methods like PRBCD. Mu et al. (2021) approximate gradients by measuring changes with small perturbations, but even under ideal conditions, their method can at best match the performance of PRBCD, which directly utilizes exact gradients. Furthermore, their approach does not scale well to graphs with even a few thousand nodes.

## A.1    Randomized smoothing-based certificates

A robustness certificate guarantees that the prediction of the classifier remains the same within a specified threat model. For any black-box model, one way to obtain such a guarantee is through randomized smoothing. A smoothing scheme $\xi$ is a random function mapping an input $\boldsymbol{x}$ to a nearby point $\boldsymbol{x}'$ (e.g. additive isotropic Gaussian noise $\boldsymbol{x}' = \xi(\boldsymbol{x}) = \boldsymbol{x} + \boldsymbol{\epsilon}$, where $\boldsymbol{\epsilon} \sim \mathcal{N}(\boldsymbol{0}, \sigma^2 \boldsymbol{I})$ for images). The smooth classifier is defined as the convolution of the smoothing scheme and the black-box classifier $g(\boldsymbol{x}) = \Pr[f(\boldsymbol{x} + \boldsymbol{\epsilon}) = y]$ – majority vote or the probability that the classifier predicts the top class for randomized $\boldsymbol{x}' \sim \xi(\boldsymbol{x})$. Regardless of the baseline classifier $f$, smooth classifier $g$ changes slowly around $\boldsymbol{x}$ and allows us to bound the worst-case minimum of the smooth prediction probability within $\mathcal{B}$. For a radius around $\boldsymbol{x}$ in which the minimum $g(\tilde{\boldsymbol{x}})$ remains above $0.5$, we can certify that the smooth model returns the same label (see § D for further details). In many smoothing schemes, exact computation of the smooth classifier is intractable. The probabilistic computation of it is also expensive as it involves many Monte-Carlo (MC) samples and later accounting for finite sample correction.

## A.2    Conformal prediction

Instead of label prediction, conformal prediction (CP) returns prediction sets that are guaranteed to include the true label with adjustable $1 - \alpha$ probability. This post-hoc method treats the model as a black-box and requires only a calibration set of labeled points whose labels were not used during model's training. CP is applicable in both inductive and transductive Graph Neural Networks (GNNs) under the assumption of node-exchangeability (Zargarbashi and Bojchevski, 2024). To compute

prediction sets we need to compute a quantile from the set of true calibration conformity scores and compare the scores (e.g. softmaxes) of the test node to the quantile threshold. For i.i.d. data (e.g. images), after computation of the quantile, the task of decreasing the softmax score towards 0 aligns with the goal of decreasing the same value below a conformal threshold (which is by definition above 0). In graphs however this task is more complicated since calibration and test nodes communicate with message passing.

## B  ISSUES WITH GRADIENT-BASED METHODS

To motivate the introduction of a search-based method, we first need to understand the shortcomings of using gradients for optimizing the discrete space of the adjacency matrix. Therefore, we study how the margin loss $L_{\text{margin}}$ changes when perturbing the adjacency matrix $A$ by flipping edges. The perturbation is defined as

$$\Delta A = e_i e_j^\top \Delta_{ij} + e_u e_v^\top \Delta_{uv},$$

where $e_i$ is the $i$-th canonical basis vector, and $\Delta_{ij}, \Delta_{uv} \in \{-1, +1\}$ denote edge additions or removals. This formulation allows us to examine the combined effect of flipping two edges simultaneously. To analyze these effects, we introduce a continuous interpolation parameter $\alpha \in [0, 1]$, and compute $L_{\text{margin}}(A + \alpha \Delta A)$. This corresponds to partially adding or removing the selected edges, giving a smooth trajectory from the original graph ($\alpha = 0$) to the fully perturbed graph ($\alpha = 1$). By searching over edge pairs, we obtain the loss landscape associated with individual and joint edge flips. Finally, we filter out those edge pairs that exhibit *non-additive behavior*: cases where flipping both edges together leads to a qualitatively different outcome compared to flipping either edge individually.

We highlight two main problems with gradient-based methods. First, the gradient is a local measure, since it quantifies the behavior of the function under infinitesimal changes. However, we are interested in the behavior of the function when flipping an edge in the discrete space $\{0, 1\}$, e.g. from 0 to 1. So, flipping an edge could increase the loss even though the gradient suggests that the loss would decrease (and the other way around). This issue was also discussed and illustrated in Zügner et al. (2018) (see their Fig. 4). Second, even if we assume that the gradient correctly indicates the effect on the loss, it still only reflects the impact of individual changes and ignores the effects of interactions between edges. There are cases where flipping each individual edge would suggest a certain direction of change in the loss (e.g. increase), but flipping both edges together would reverse the direction (e.g. decrease).

We designed an experiment to demonstrate that these phenomena are not rare. Since the search space is very large, we start with a specific node and then randomly sample towards the other side of its edges. Specifically, we are looking for the edges $(i, j)$ and $(i, v)$ with $u = i$. We chose this approach because it ensures that the changed edge remains within the first-hop neighborhood of the node. Since our GCN is a two-layer network, the probability that this edge interacts with the two-hop neighborhood of the graph also increases. We found several cases of these two events for each node in the `CoraML` dataset. Fig. 7 visualizes a random subset of these cases based on `Tanh-Margin` loss. The same phenomena also occurs with `Cross-Entropy` loss. Fig. 8 visualizes a random subset of these cases using `Cross-Entropy` loss.

## C  SUPPLEMENTARY EXPERIMENTS

**Transductive Setting.** In § 5, we argued that transductive setup carries a false sense of robustness. In this setup, trivial robustness can be gained just by memorization of the clean graph (Gosch et al., 2024). For completeness, here we report the results in the transductive setup as well. As in Table 6 EvA outperforms SOTA consistent with other experiments in the inductive setup.

**Stratified sampling.** Although unrealistic, in Table 7, we compare attacks in case the models are trained train/val/test sampled with the same number of nodes across different classes. Consistent with other results, EvA shows to be better here as well.

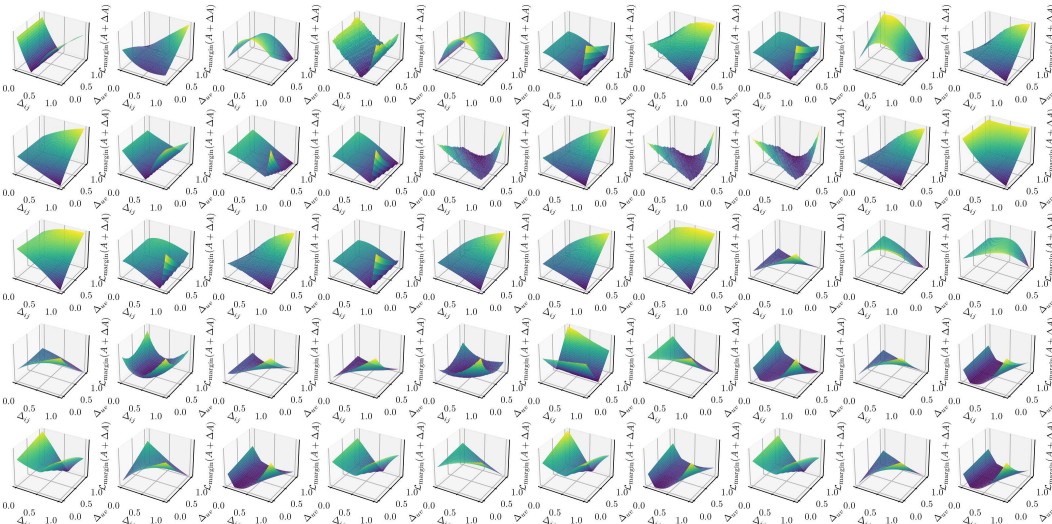

Figure 7: In some cases the gradient fails to measure the effect of flipping an edge on the `Tanh-Margin` loss. Flipping edges individually vs. jointly has a different effect on loss.

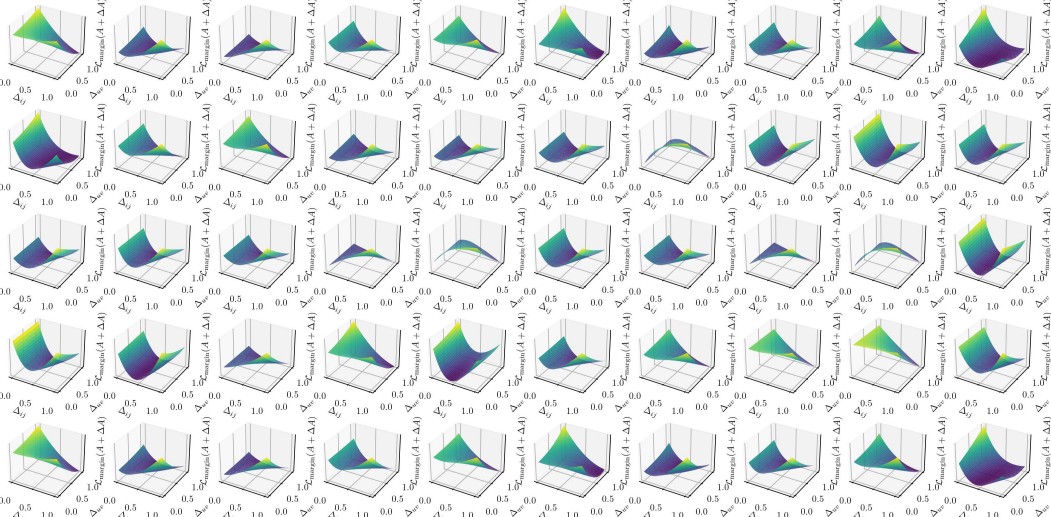

Figure 8: In some cases the gradient fails to measure the effect of flipping an edge on the `Cross-Entropy` loss. Flipping edges individually vs. jointly has a different effect on loss.

## C.1 INDUCTIVE SETTING, NON-STRATIFIED SAMPLING

**Vanilla models.** Here, we present additional results specifically for the inductive setting. With the discussion in § 5 our main experimental setup is for inductive GNNs. In this section, we detail the effectiveness of our method compared to other approaches. We show the results for `CoraML`, `Pubmed`, and `Citeseer` datasets and vanilla models in tables 8, 9, 10, 11, 12, and 13. For each dataset we compare our attack with SOTA on four models: GCN, GAT, APPNP, and GPRGNN. We further compare the local variant of EvA with LRBCD under local-degree constraint for GCN, and GPRGNN.

**Models with robust training.** Table 14 compares EvA and SOTA for models training with robust training. During training of these models, we use an adversarial attack at each step to attack $\mathcal{G}_{\mathrm{tr}}$, and then we retrain the model on the adversarially perturbed graph. The robust budget ($\epsilon_{\mathrm{robust}}$) for adversarial attack during training was $0.2$. This process repeats in each epoch of training until the model converges. Gosch et al. (2023) shows that models with adversarial and self training carry a

Table 6: Classification accuracy (%) on the `CoraML` dataset in the transductive setting under adversarial attacks. Results are reported for two GNN models subjected to three different attack methods across varying perturbation budgets $\epsilon$. Training, validation, and test sets are non-stratified.

| Model | Attack | $\epsilon$ | | | | | |
|-------|--------|------|------|------|------|------|------|
| | | 0.01 | 0.02 | 0.05 | 0.10 | 0.15 | 0.20 |
| GCN | LRBCD | $79.88_{\pm1.52}$ | $77.66_{\pm1.84}$ | $72.61_{\pm1.38}$ | $65.84_{\pm1.44}$ | $60.80_{\pm1.82}$ | $56.86_{\pm1.92}$ |
| | PRBCD | $79.48_{\pm1.70}$ | $77.26_{\pm1.59}$ | $71.85_{\pm1.80}$ | $65.49_{\pm1.73}$ | $60.49_{\pm2.29}$ | $\mathbf{56.07}_{\pm2.36}$ |
| | EvA | $\mathbf{77.38}_{\pm1.87}$ | $\mathbf{74.12}_{\pm1.99}$ | $\mathbf{65.68}_{\pm1.97}$ | $60.49_{\pm2.31}$ | $58.72_{\pm2.57}$ | $57.41_{\pm2.51}$ |
| GPRGNN | LRBCD | $79.24_{\pm3.04}$ | $77.09_{\pm3.25}$ | $71.66_{\pm4.43}$ | $62.74_{\pm7.28}$ | $55.01_{\pm11.96}$ | $\mathbf{49.04}_{\pm15.66}$ |
| | PRBCD | $78.67_{\pm3.20}$ | $75.86_{\pm3.69}$ | $69.83_{\pm5.09}$ | $62.15_{\pm7.47}$ | $55.15_{\pm10.17}$ | $50.13_{\pm12.63}$ |
| | EvA | $\mathbf{76.53}_{\pm3.46}$ | $\mathbf{72.74}_{\pm4.53}$ | $\mathbf{63.78}_{\pm6.19}$ | $\mathbf{56.83}_{\pm9.98}$ | $\mathbf{53.00}_{\pm13.01}$ | $50.04_{\pm15.53}$ |

Table 7: Classification accuracy (%) on the `CoraML` dataset in the inductive setting under adversarial attacks. Results are reported for two GNN models subjected to three different attack methods across varying perturbation budgets $\epsilon$. Training, validation, and test sets are stratified.

| Model | Attack | $\epsilon$ | | | | | |
|-------|--------|------|------|------|------|------|------|
| | | 0.01 | 0.02 | 0.05 | 0.10 | 0.15 | 0.20 |
| GCN | LRBCD | $80.00_{\pm2.70}$ | $77.43_{\pm2.58}$ | $71.43_{\pm2.72}$ | $65.50_{\pm3.91}$ | $61.21_{\pm4.25}$ | $57.64_{\pm4.54}$ |
| | PRBCD | $78.71_{\pm2.80}$ | $75.29_{\pm3.42}$ | $67.86_{\pm3.43}$ | $59.50_{\pm3.50}$ | $53.00_{\pm4.14}$ | $48.43_{\pm3.73}$ |
| | EvA | $\mathbf{77.00}_{\pm2.86}$ | $\mathbf{71.36}_{\pm3.29}$ | $\mathbf{54.36}_{\pm4.73}$ | $\mathbf{44.29}_{\pm3.28}$ | $\mathbf{40.86}_{\pm3.73}$ | $\mathbf{37.29}_{\pm3.73}$ |
| GPRGNN | LRBCD | $76.21_{\pm7.94}$ | $73.14_{\pm7.88}$ | $66.07_{\pm11.50}$ | $60.79_{\pm11.74}$ | $56.43_{\pm12.43}$ | $53.36_{\pm12.90}$ |
| | PRBCD | $74.36_{\pm9.60}$ | $70.71_{\pm9.96}$ | $63.79_{\pm10.44}$ | $56.14_{\pm11.14}$ | $49.29_{\pm11.44}$ | $45.07_{\pm11.04}$ |
| | EvA | $\mathbf{71.86}_{\pm10.37}$ | $\mathbf{65.00}_{\pm11.44}$ | $\mathbf{50.14}_{\pm11.44}$ | $\mathbf{41.79}_{\pm14.02}$ | $\mathbf{37.21}_{\pm14.58}$ | $\mathbf{35.21}_{\pm15.67}$ |

false sense of robustness in transductive setup, therefore same as other experiments we evaluate in inductive setup. Similar to vanilla models, EvA outperforms all previous attacks.

## C.2 COMPARING OUR METHOD WITH OTHER ATTACKS

In the main paper, we mainly focus on comparing EvA with PRBCD since it is the SOTA white-box attack. Here, we provide a more comprehensive comparison with other attacks including DICE (Zügner et al., 2018), FGSM (Xu et al., 2019), PGD (Zügner et al., 2018), and GRBCD (Geisler et al., 2023). We compare all these methods on the `CoraML` and `Pubmed` dataset in Fig. 1, Table 15. As shown, EvA outperforms all other attacks by a significant margin.

## C.3 OGBN-ARXIV DATASET

To show the scalability of our attack on larger graphs, we present results on the large `Ogbn-Arxiv` dataset. We compare EvA and PRBCD on the same setup as other experiments. In the main paper and § D.4, we used arxiv with divide and concur and show it can outperform PRBCD. Further we propose two other setups where the attack is more realistic: (i) Smaller perturbation budgets: perturbing the `Ogbn-Arxiv` dataset with the same budget as a smaller graph like `CoraML` is unrealistic. Therefore we can decrease $\epsilon$, by one order of magnitude and evaluate both methods on $\epsilon \in \{0.1\%, 0.5\%, 1.0\%\}$. The results for these budgets are summarized in Table 16. (ii) Simialrly another realistic setup is that on a large graph, the adversary can access a smaller subset of control nodes (e.g. 1000 nodes) with the objective to perturb a set of target nodes. As an example in a social network, an attacker could purchase 1,000 user accounts and use them to influence the performance of other subgroups. Here, we randomly sampled 1,000 nodes as control and 1,500 nodes as target for 5 rounds. We compared EvA without D&C and PRBCD and reported the average results in Table 17. Our method outperforms PRBCD in this scenario as well.

Table 8: Classification accuracy (%) on the `CoraML` dataset in the inductive setting under adversarial attacks. Results are reported for four GNN models subjected to two different attack methods across varying perturbation budgets $\epsilon$. Training, validation, and test sets are non-stratified.

| Model | Attack | $\epsilon$ | | | | | |
|---|---|---|---|---|---|---|---|
| | | 0.01 | 0.02 | 0.05 | 0.10 | 0.15 | 0.20 |
| APPNP | EvA | $\mathbf{76.65}_{\pm 1.32}$ | $\mathbf{71.03}_{\pm 1.44}$ | $\mathbf{56.51}_{\pm 1.60}$ | $\mathbf{49.32}_{\pm 1.84}$ | $\mathbf{44.77}_{\pm 2.04}$ | $\mathbf{41.42}_{\pm 1.41}$ |
| | PRBCD | $78.65_{\pm 0.99}$ | $75.30_{\pm 1.27}$ | $68.75_{\pm 1.22}$ | $61.57_{\pm 1.65}$ | $55.44_{\pm 1.58}$ | $49.96_{\pm 2.42}$ |
| GAT | EvA | $\mathbf{64.20}_{\pm 1.89}$ | $\mathbf{58.51}_{\pm 2.45}$ | $\mathbf{40.99}_{\pm 1.60}$ | $\mathbf{15.30}_{\pm 4.47}$ | $\mathbf{9.40}_{\pm 6.83}$ | $\mathbf{8.11}_{\pm 6.65}$ |
| | PRBCD | $70.07_{\pm 2.82}$ | $66.55_{\pm 2.21}$ | $58.58_{\pm 3.33}$ | $49.61_{\pm 6.55}$ | $39.86_{\pm 6.78}$ | $36.94_{\pm 7.09}$ |
| GCN | EvA | $\mathbf{74.80}_{\pm 1.50}$ | $\mathbf{68.97}_{\pm 1.58}$ | $\mathbf{52.95}_{\pm 1.91}$ | $\mathbf{41.99}_{\pm 2.06}$ | $\mathbf{37.65}_{\pm 2.74}$ | $\mathbf{35.37}_{\pm 2.38}$ |
| | PRBCD | $76.44_{\pm 1.64}$ | $73.17_{\pm 1.39}$ | $66.48_{\pm 2.13}$ | $58.51_{\pm 1.77}$ | $52.67_{\pm 2.09}$ | $47.19_{\pm 2.02}$ |
| GPRGNN | EvA | $\mathbf{72.53}_{\pm 4.11}$ | $\mathbf{66.83}_{\pm 4.54}$ | $\mathbf{51.53}_{\pm 5.57}$ | $\mathbf{42.21}_{\pm 8.52}$ | $\mathbf{37.01}_{\pm 9.83}$ | $\mathbf{34.52}_{\pm 9.83}$ |
| | PRBCD | $74.95_{\pm 3.08}$ | $71.67_{\pm 2.76}$ | $64.84_{\pm 4.18}$ | $57.94_{\pm 4.55}$ | $53.24_{\pm 5.20}$ | $48.68_{\pm 6.52}$ |

Table 9: Classification accuracy (%) on the `CoraML` dataset in the inductive setting under adversarial attacks. Results are reported for two GNN models subjected to five different attack methods across varying perturbation budgets $\epsilon$. Training, validation, and test sets are non-stratified.

| Model | Attack | $\epsilon$ | | | | | |
|---|---|---|---|---|---|---|---|
| | | 0.01 | 0.02 | 0.05 | 0.10 | 0.15 | 0.20 |
| GCN | EvA | $\mathbf{74.80}_{\pm 1.50}$ | $\mathbf{68.97}_{\pm 1.58}$ | $\mathbf{52.95}_{\pm 1.91}$ | $\mathbf{41.99}_{\pm 2.06}$ | $\mathbf{37.65}_{\pm 2.74}$ | $\mathbf{35.37}_{\pm 2.38}$ |
| | EvaLocal | $75.09_{\pm 1.73}$ | $69.82_{\pm 1.96}$ | $60.21_{\pm 2.04}$ | $56.09_{\pm 1.93}$ | $54.16_{\pm 2.48}$ | $52.88_{\pm 1.79}$ |
| | LRBCD | $78.51_{\pm 1.56}$ | $75.94_{\pm 1.54}$ | $71.10_{\pm 1.16}$ | $64.41_{\pm 1.65}$ | $60.14_{\pm 1.73}$ | $57.37_{\pm 1.45}$ |
| | PRBCD | $76.44_{\pm 1.64}$ | $73.17_{\pm 1.39}$ | $66.48_{\pm 2.13}$ | $58.51_{\pm 1.77}$ | $52.67_{\pm 2.09}$ | $47.19_{\pm 2.02}$ |
| | PGA | $79.58_{\pm 1.61}$ | $76.92_{\pm 1.73}$ | $70.94_{\pm 1.89}$ | $64.62_{\pm 1.92}$ | $60.46_{\pm 2.25}$ | $57.54_{\pm 2.46}$ |
| GPRGNN | EvA | $\mathbf{72.53}_{\pm 4.11}$ | $\mathbf{66.83}_{\pm 4.54}$ | $\mathbf{51.53}_{\pm 5.57}$ | $\mathbf{42.21}_{\pm 8.52}$ | $\mathbf{37.01}_{\pm 9.83}$ | $\mathbf{34.52}_{\pm 9.83}$ |
| | EvaLocal | $73.31_{\pm 3.30}$ | $67.26_{\pm 4.17}$ | $58.29_{\pm 7.96}$ | $53.38_{\pm 11.42}$ | $51.10_{\pm 12.66}$ | $49.96_{\pm 13.63}$ |
| | LRBCD | $77.51_{\pm 1.81}$ | $74.80_{\pm 1.41}$ | $68.83_{\pm 1.90}$ | $62.56_{\pm 1.71}$ | $59.07_{\pm 1.53}$ | $55.66_{\pm 1.71}$ |
| | PRBCD | $74.95_{\pm 3.08}$ | $71.67_{\pm 2.76}$ | $64.84_{\pm 4.18}$ | $57.94_{\pm 4.55}$ | $53.24_{\pm 5.20}$ | $48.68_{\pm 6.52}$ |
| | PGA | $78.55_{\pm 3.03}$ | $75.33_{\pm 3.69}$ | $68.63_{\pm 5.11}$ | $61.55_{\pm 6.97}$ | $56.60_{\pm 8.52}$ | $54.91_{\pm 7.46}$ |

Table 16: Accuracy (%) on the `Ogbn-Arxiv` dataset under EvA and PRBCD across varying perturbation budgets $\epsilon$.

| Attack | Clean | 0.1% | 0.5% | 1% |
|---|---|---|---|---|
| PRBCD | 70.53 | 69.83 | 68.64 | **66.27** |
| EvA | 70.53 | **69.21** | **67.59** | 66.86 |

Table 17: Accuracy (%) on the `Ogbn-Arxiv` dataset under EvA and PRBCD across varying perturbation budgets $\epsilon$ using control nodes.

| Attack | Clean | 1% | 5% |
|---|---|---|---|
| PRBCD | | 64.89 | 54.7 |
| EvA | | **59.3** | **53.92** |

In addition to both realistic cases, we compared EvA (with divide and conquer) to PRBCD in the same setup as we evaluated for other datasets. The summarized result is illustrated in Fig. 5.

## C.4 Comparison with (Dai et al., 2018b)

(Dai et al., 2018b) proposed a practical black-box attack (PBA), dividing it into PBA-C (with access to logits - continuous) and PBA-D (access only to the labels - discrete). As stated in (Dai et al., 2018b), a genetic algorithm for global attacks requires PBA-C because it relies on logits, with the fitness function being the negative log-likelihood. We demonstrate that EvA not only eliminates the need for logits but also performs even better by directly optimizing for accuracy rather than using log-likelihood. To compare our method with (Dai et al., 2018b), we modified the algorithm's

Table 10: Classification accuracy (%) on the `PubMed` dataset in the inductive setting under adversarial attacks. Results are reported for four GNN models subjected to two different attack methods across varying perturbation budgets $\epsilon$. Training, validation, and test sets are non-stratified.

| Model | Attack | $\epsilon$ | | | | | |
|---|---|---|---|---|---|---|---|
| | | 0.01 | 0.02 | 0.05 | 0.10 | 0.15 | 0.20 |
| APPNP | EvA | $\mathbf{73.85}_{\pm2.35}$ | $\mathbf{69.64}_{\pm2.16}$ | $\mathbf{57.07}_{\pm2.32}$ | $47.03_{\pm2.18}$ | $\mathbf{43.94}_{\pm1.83}$ | $\mathbf{41.93}_{\pm2.18}$ |
| | PRBCD | $75.54_{\pm2.34}$ | $72.44_{\pm2.28}$ | $65.14_{\pm2.26}$ | $57.15_{\pm2.59}$ | $51.04_{\pm2.79}$ | $45.75_{\pm2.60}$ |
| GAT | EvA | $\mathbf{69.15}_{\pm1.83}$ | $\mathbf{64.62}_{\pm1.81}$ | $\mathbf{52.17}_{\pm1.71}$ | $\mathbf{33.62}_{\pm2.05}$ | $\mathbf{26.62}_{\pm3.74}$ | $\mathbf{24.21}_{\pm4.27}$ |
| | PRBCD | $71.33_{\pm1.53}$ | $67.78_{\pm1.79}$ | $59.73_{\pm2.10}$ | $49.87_{\pm1.36}$ | $42.04_{\pm1.57}$ | $35.94_{\pm1.66}$ |
| GCN | EvA | $\mathbf{72.60}_{\pm2.19}$ | $\mathbf{68.35}_{\pm2.41}$ | $\mathbf{56.15}_{\pm1.92}$ | $\mathbf{42.93}_{\pm2.64}$ | $\mathbf{40.46}_{\pm2.76}$ | $\mathbf{39.11}_{\pm2.98}$ |
| | PRBCD | $74.99_{\pm1.99}$ | $71.90_{\pm2.03}$ | $64.16_{\pm2.32}$ | $55.54_{\pm2.79}$ | $49.32_{\pm2.66}$ | $43.90_{\pm3.09}$ |
| GPRGNN | EvA | $\mathbf{72.01}_{\pm4.18}$ | $\mathbf{67.61}_{\pm4.28}$ | $\mathbf{55.95}_{\pm4.32}$ | $\mathbf{51.49}_{\pm7.51}$ | $\mathbf{49.18}_{\pm7.83}$ | $\mathbf{42.39}_{\pm9.63}$ |
| | PRBCD | $74.37_{\pm3.40}$ | $71.66_{\pm3.55}$ | $64.51_{\pm4.94}$ | $56.21_{\pm6.46}$ | $50.26_{\pm7.41}$ | $45.81_{\pm8.47}$ |

Table 11: Classification accuracy (%) on the `PubMed` dataset in the inductive setting under adversarial attacks. Results are reported for two GNN models subjected to four different attack methods across varying perturbation budgets $\epsilon$. Training, validation, and test sets are non-stratified.

| Model | Attack | $\epsilon$ | | | | | |
|---|---|---|---|---|---|---|---|
| | | 0.01 | 0.02 | 0.05 | 0.10 | 0.15 | 0.20 |
| GCN | EvA | $72.60_{\pm2.18}$ | $68.35_{\pm2.41}$ | $56.15_{\pm1.92}$ | $42.93_{\pm2.64}$ | $40.46_{\pm2.76}$ | $39.11_{\pm2.98}$ |
| | EvaLocal | $74.12_{\pm2.19}$ | $69.99_{\pm2.04}$ | $63.43_{\pm2.76}$ | $61.51_{\pm2.64}$ | $61.01_{\pm2.79}$ | $60.54_{\pm2.64}$ |
| | LRBCD | $74.89_{\pm2.04}$ | $71.48_{\pm2.49}$ | $65.68_{\pm2.90}$ | $60.24_{\pm3.15}$ | $56.81_{\pm3.02}$ | $54.07_{\pm2.99}$ |
| | PRBCD | $74.99_{\pm1.99}$ | $71.90_{\pm2.03}$ | $64.16_{\pm2.32}$ | $55.54_{\pm2.79}$ | $49.32_{\pm2.66}$ | $43.90_{\pm3.09}$ |
| GPRGNN | EvA | $72.01_{\pm4.18}$ | $67.61_{\pm4.28}$ | $55.95_{\pm4.32}$ | $51.49_{\pm7.51}$ | $49.18_{\pm7.83}$ | $42.39_{\pm9.63}$ |
| | EvALocal | $73.01_{\pm4.18}$ | $69.10_{\pm3.83}$ | $62.77_{\pm6.59}$ | $60.51_{\pm8.10}$ | $59.72_{\pm8.91}$ | $59.31_{\pm9.50}$ |
| | LRBCD | $74.50_{\pm3.66}$ | $71.57_{\pm4.10}$ | $65.88_{\pm6.12}$ | $60.33_{\pm5.70}$ | $56.75_{\pm7.74}$ | $53.75_{\pm8.18}$ |
| | PRBCD | $74.37_{\pm3.40}$ | $71.66_{\pm3.55}$ | $64.51_{\pm4.94}$ | $56.21_{\pm6.46}$ | $50.26_{\pm7.41}$ | $45.81_{\pm8.47}$ |

fitness function and mutation mechanism to replicate the results reported in (Dai et al., 2018b). This implementation retains scalability benefits, as it is also built upon our sparse encoded representation. Note here we re-implement Dai et al. (2018b) in our sparse and parallelized framework. Their original implementation uses dense adjacency matrices and sequential evaluation and would achieve a significantly worse result within the same memory/run-time constraint. Even with our efficient re-implementation Dai et al. (2018b) is significantly worse than ours. Table 18 provides the results for the `CoraML` dataset using the GCN architecture. EvA also significantly outperforms (Dai et al., 2018b).

Additionally, since our method is independent of gradients, we established the first attack on conformal prediction and certification. For conformal prediction, we attack coverage and set size where the latter criteria are not yet explored (to the best of our knowledge). Attacks tending to decrease certificate effectiveness are also under-explored in GNNs. In this work, we aim to achieve both attack on certified accuracy and certified ratio.

## D   TECHNICAL DETAILS OF EVA

**Rigorous definition for components in EvA.** We define a genetic solver with four main components. (i) Population: a set of feasible answers to the problem that gradually improve over iterations. Here, each candidate is a perturbation to the original graph, a vector of indices at which an edge will flip; formally $s_i \in [\frac{n}{2}(n-1)]^\delta$. Indices are calculated via a mapping $\Pi : [n]^2 \mapsto [\frac{n}{2}(n-1)]$ that is an enumeration on the upper triangle of the $n \times n$ adjacency matrix (see § D). The corresponding

Table 12: Classification accuracy (%) on the `Citeseer` dataset in the inductive setting under adversarial attacks. Results are reported for four GNN models subjected to two different attack methods across varying perturbation budgets $\epsilon$. Training, validation, and test sets are non-stratified.

| Model | Attack | $\epsilon$ | | | | | |
|---|---|---|---|---|---|---|---|
| | | 0.01 | 0.02 | 0.05 | 0.10 | 0.15 | 0.20 |
| APPNP | EvA | $\mathbf{86.90}_{\pm 1.11}$ | $\mathbf{83.93}_{\pm 0.94}$ | $\mathbf{74.29}_{\pm 0.88}$ | $\mathbf{65.00}_{\pm 1.15}$ | $\mathbf{59.76}_{\pm 2.33}$ | $\mathbf{54.76}_{\pm 1.19}$ |
| | PRBCD | $87.26_{\pm 0.90}$ | $85.48_{\pm 1.49}$ | $81.79_{\pm 1.08}$ | $76.55_{\pm 0.68}$ | $72.44_{\pm 1.66}$ | $69.29_{\pm 1.69}$ |
| GAT | EvA | $\mathbf{81.54}_{\pm 1.39}$ | $\mathbf{76.78}_{\pm 2.22}$ | $\mathbf{67.14}_{\pm 3.65}$ | $\mathbf{51.19}_{\pm 4.21}$ | $\mathbf{37.74}_{\pm 4.94}$ | $\mathbf{27.62}_{\pm 9.49}$ |
| | PRBCD | $84.52_{\pm 2.27}$ | $82.62_{\pm 2.20}$ | $76.55_{\pm 5.59}$ | $70.00_{\pm 6.09}$ | $67.02_{\pm 4.27}$ | $63.15_{\pm 4.19}$ |
| GCN | EvA | $\mathbf{86.67}_{\pm 1.71}$ | $\mathbf{82.86}_{\pm 2.12}$ | $\mathbf{72.74}_{\pm 2.74}$ | $\mathbf{58.33}_{\pm 3.01}$ | $\mathbf{49.76}_{\pm 3.22}$ | $\mathbf{44.29}_{\pm 3.33}$ |
| | PRBCD | $87.38_{\pm 1.81}$ | $85.83_{\pm 2.43}$ | $80.95_{\pm 2.06}$ | $74.29_{\pm 4.22}$ | $69.76_{\pm 4.34}$ | $67.62_{\pm 4.96}$ |
| GPRGNN | EvA | $\mathbf{87.26}_{\pm 2.75}$ | $\mathbf{83.81}_{\pm 2.50}$ | $\mathbf{73.45}_{\pm 3.17}$ | $61.43_{\pm 4.66}$ | $\mathbf{55.48}_{\pm 3.84}$ | $\mathbf{50.12}_{\pm 4.86}$ |
| | PRBCD | $88.45_{\pm 2.29}$ | $86.31_{\pm 2.45}$ | $82.02_{\pm 2.61}$ | $77.14_{\pm 2.84}$ | $73.93_{\pm 3.89}$ | $69.64_{\pm 3.47}$ |

Table 13: Classification accuracy (%) on the `Citeseer` dataset in the inductive setting under adversarial attacks. Results are reported for two GNN models subjected to four different attack methods across varying perturbation budgets $\epsilon$. Training, validation, and test sets are non-stratified.

| Model | Attack | $\epsilon$ | | | | | |
|---|---|---|---|---|---|---|---|
| | | 0.01 | 0.02 | 0.05 | 0.10 | 0.15 | 0.20 |
| GCN | EvA | $\mathbf{86.67}_{\pm 1.71}$ | $\mathbf{82.86}_{\pm 2.12}$ | $\mathbf{72.74}_{\pm 2.74}$ | $\mathbf{58.33}_{\pm 3.01}$ | $\mathbf{49.76}_{\pm 3.22}$ | $\mathbf{44.29}_{\pm 3.33}$ |
| | EvaLocal | $87.38_{\pm 1.65}$ | $83.57_{\pm 2.17}$ | $78.21_{\pm 3.17}$ | $76.43_{\pm 2.62}$ | $75.00_{\pm 3.23}$ | $74.52_{\pm 3.16}$ |
| | LRBCD | $88.45_{\pm 2.17}$ | $86.43_{\pm 2.71}$ | $83.69_{\pm 2.48}$ | $80.12_{\pm 3.30}$ | $78.45_{\pm 3.89}$ | $75.36_{\pm 4.81}$ |
| | PRBCD | $87.38_{\pm 1.81}$ | $85.83_{\pm 2.43}$ | $80.95_{\pm 2.06}$ | $74.29_{\pm 4.22}$ | $69.76_{\pm 4.34}$ | $67.62_{\pm 4.96}$ |
| GPRGNN | EvA | $\mathbf{87.26}_{\pm 2.75}$ | $\mathbf{83.81}_{\pm 2.50}$ | $\mathbf{73.45}_{\pm 3.17}$ | $61.43_{\pm 4.66}$ | $\mathbf{55.48}_{\pm 3.84}$ | $\mathbf{50.12}_{\pm 4.86}$ |
| | EvaLocal | $87.50_{\pm 2.27}$ | $84.29_{\pm 2.04}$ | $80.48_{\pm 3.96}$ | $77.86_{\pm 4.56}$ | $76.43_{\pm 5.06}$ | $75.12_{\pm 6.57}$ |
| | LRBCD | $89.76_{\pm 2.50}$ | $87.98_{\pm 2.48}$ | $85.12_{\pm 2.76}$ | $81.90_{\pm 2.83}$ | $79.64_{\pm 4.08}$ | $78.45_{\pm 4.92}$ |
| | PRBCD | $88.45_{\pm 2.29}$ | $86.31_{\pm 2.45}$ | $82.02_{\pm 2.61}$ | $77.14_{\pm 2.84}$ | $73.93_{\pm 3.89}$ | $69.64_{\pm 3.47}$ |

perturbation matrix $\boldsymbol{P}_i$ is simply defined as $\boldsymbol{P}_i[p_t, q_t] = \boldsymbol{P}_i[q_t, p_t] = 1$ where $(p_t, q_t) = \Pi^{-1}(\boldsymbol{s}_i[t])$ for every index $t$. The initial population is selected randomly. (ii) Fitness: is a notion of how close to optimal each candidate is. For any loss function $\mathcal{L}$ we define the fitness function $\mathrm{fit} : [\frac{n}{2}(n-1)]^\delta \mapsto \mathbb{R}$, as $\mathrm{fit}(\boldsymbol{s}_i) = \mathcal{L}(\boldsymbol{X}, \boldsymbol{A} \oplus \boldsymbol{P}_i, \boldsymbol{y})$. Regardless of differentiability, as long as the loss function has enough sensitivity to contrast between various individuals, we use it directly to compute the fitness (special case in § 4). (iii) Crossover: is an operation to generate new population candidate via combining two existing ones. The (single joint) crossover operation at joint $j$ defines a new candidate vector $\boldsymbol{s}_{\mathrm{new}} = \mathrm{cross}_j(\boldsymbol{s}_1, \boldsymbol{s}_2) := \boldsymbol{s}_1[: j] \bullet \boldsymbol{s}_2[j+1 :]$ where $\bullet$ is the concatenation of two vectors. Crossover operation with $k_{\mathrm{cross}} > 1$ joints is defined recursively in the order of joints. The number of crossovers is a hyperparameter (see § E), and their locations are chosen randomly in the range of candidates' length. The candidates for cross-over are chosen through a "tournament". In each tournament, $n_{\mathrm{tour}}$ random candidates are compared, and the parent candidates are selected based on their fitness. This process repeats for $t$ generations. (iv) Mutation: introduces further exploration to the new population. The function $\mathrm{mutate} : [\frac{n}{2}(n-1)]^\delta \mapsto [\frac{n}{2}(n-1)]^\delta$ is a random mapping of a candidate to another. A simple example of mutation is to change each index with some mutation probability $p$ to any random index in the range. We propose better mutation operators later.

**Mapping function: enumeration over $\boldsymbol{A}$.** For enumerating over $\boldsymbol{A}$, instead of using the row and column indices of the node to select, we introduced indexing. For a directed graph, the indexing starts from 0 to $n^2 - 1$. However, in an undirected graph, we only need the upper triangular part of the matrix $\boldsymbol{A}$. To achieve this, we use the following algebraic solution to find the row and column of

Table 14: Classification accuracy (%) on the `CoraML` dataset in the inductive setting under adversarial attacks. Results are reported for two GNN models with or without adversarial training subjected to three different attack methods across varying perturbation budgets $\epsilon$. Training, validation, and test sets are stratified.

| Model | Adv. Tr. | Attack | $\epsilon$ | | | | | |
|---|---|---|---|---|---|---|---|---|
| | | | 0.01 | 0.02 | 0.05 | 0.10 | 0.15 | 0.20 |
| GCN | None | LRBCD | $78.51_{\pm1.56}$ | $75.94_{\pm1.54}$ | $71.10_{\pm1.16}$ | $64.41_{\pm1.65}$ | $60.14_{\pm1.73}$ | $57.37_{\pm1.45}$ |
| | | PRBCD | $76.44_{\pm1.64}$ | $73.17_{\pm1.39}$ | $66.48_{\pm2.13}$ | $58.51_{\pm1.77}$ | $52.67_{\pm2.09}$ | $47.19_{\pm2.02}$ |
| | | EvA | $74.80_{\pm1.50}$ | $68.97_{\pm1.58}$ | $52.95_{\pm1.91}$ | $41.99_{\pm2.06}$ | $37.65_{\pm2.74}$ | $35.37_{\pm2.38}$ |
| | LRBCD | LRBCD | $79.64_{\pm1.77}$ | $77.51_{\pm2.41}$ | $73.10_{\pm1.54}$ | $68.19_{\pm1.11}$ | $64.84_{\pm1.92}$ | $62.35_{\pm3.00}$ |
| | | PRBCD | $78.79_{\pm1.88}$ | $75.87_{\pm1.41}$ | $69.75_{\pm1.81}$ | $62.35_{\pm2.70}$ | $56.80_{\pm3.04}$ | $54.23_{\pm4.71}$ |
| | | EvA | $76.80_{\pm1.29}$ | $71.10_{\pm1.64}$ | $56.30_{\pm1.66}$ | $48.40_{\pm2.91}$ | $43.06_{\pm2.43}$ | $40.85_{\pm2.70}$ |
| | PRBCD | LRBCD | $80.71_{\pm1.16}$ | $77.86_{\pm0.81}$ | $73.81_{\pm0.54}$ | $69.40_{\pm0.91}$ | $66.48_{\pm1.08}$ | $63.77_{\pm1.73}$ |
| | | PRBCD | $78.93_{\pm1.27}$ | $76.30_{\pm1.27}$ | $70.25_{\pm1.74}$ | $64.06_{\pm1.83}$ | $59.50_{\pm2.84}$ | $56.58_{\pm4.53}$ |
| | | EvA | $76.80_{\pm0.77}$ | $71.53_{\pm1.65}$ | $57.44_{\pm2.13}$ | $49.11_{\pm3.05}$ | $44.70_{\pm2.96}$ | $41.92_{\pm3.32}$ |
| | EvA | LRBCD | $80.85_{\pm1.36}$ | $78.58_{\pm0.99}$ | $74.66_{\pm1.11}$ | $69.89_{\pm0.93}$ | $66.98_{\pm0.92}$ | $65.05_{\pm1.08}$ |
| | | PRBCD | $79.79_{\pm1.80}$ | $76.51_{\pm1.31}$ | $71.25_{\pm1.54}$ | $64.34_{\pm1.97}$ | $60.43_{\pm1.32}$ | $58.22_{\pm2.26}$ |
| | | EvA | $77.22_{\pm1.87}$ | $71.96_{\pm2.38}$ | $57.94_{\pm3.08}$ | $50.04_{\pm3.29}$ | $44.91_{\pm3.44}$ | $42.63_{\pm2.33}$ |
| GPRGNN | None | LRBCD | $77.51_{\pm2.81}$ | $74.80_{\pm3.08}$ | $68.83_{\pm4.20}$ | $62.56_{\pm4.69}$ | $59.07_{\pm5.98}$ | $55.66_{\pm6.99}$ |
| | | PRBCD | $74.95_{\pm3.08}$ | $71.67_{\pm2.76}$ | $64.84_{\pm4.18}$ | $57.94_{\pm4.55}$ | $53.24_{\pm5.20}$ | $48.68_{\pm6.52}$ |
| | | EvA | $72.53_{\pm4.11}$ | $66.83_{\pm4.54}$ | $51.53_{\pm5.57}$ | $42.21_{\pm8.52}$ | $37.01_{\pm9.84}$ | $34.52_{\pm9.83}$ |
| | LRBCD | LRBCD | $81.57_{\pm2.58}$ | $79.72_{\pm2.22}$ | $75.59_{\pm2.31}$ | $71.32_{\pm2.20}$ | $68.97_{\pm2.10}$ | $66.69_{\pm2.25}$ |
| | | PRBCD | $80.71_{\pm2.61}$ | $78.51_{\pm2.29}$ | $72.88_{\pm2.38}$ | $66.90_{\pm1.95}$ | $61.78_{\pm1.99}$ | $57.51_{\pm3.72}$ |
| | | EvA | $78.79_{\pm2.69}$ | $72.95_{\pm2.67}$ | $63.42_{\pm3.15}$ | $56.58_{\pm4.68}$ | $52.88_{\pm5.61}$ | $49.96_{\pm5.75}$ |
| | PRBCD | LRBCD | $80.43_{\pm2.01}$ | $78.01_{\pm1.91}$ | $73.74_{\pm1.66}$ | $69.96_{\pm2.14}$ | $67.19_{\pm2.51}$ | $64.84_{\pm3.20}$ |
| | | PRBCD | $80.21_{\pm2.43}$ | $77.30_{\pm2.63}$ | $71.53_{\pm2.67}$ | $65.12_{\pm3.21}$ | $60.07_{\pm4.10}$ | $55.37_{\pm3.85}$ |
| | | EvA | $78.79_{\pm2.45}$ | $73.10_{\pm2.54}$ | $62.85_{\pm4.93}$ | $56.94_{\pm6.64}$ | $53.74_{\pm7.65}$ | $51.60_{\pm8.10}$ |
| | EvA | LRBCD | $79.64_{\pm0.89}$ | $76.44_{\pm0.68}$ | $72.95_{\pm1.04}$ | $69.04_{\pm1.26}$ | $67.05_{\pm1.46}$ | $65.48_{\pm1.88}$ |
| | | PRBCD | $78.51_{\pm0.60}$ | $75.87_{\pm1.32}$ | $70.32_{\pm0.89}$ | $64.91_{\pm1.14}$ | $59.57_{\pm1.75}$ | $56.16_{\pm1.62}$ |
| | | EvA | $76.51_{\pm0.44}$ | $70.96_{\pm0.41}$ | $60.85_{\pm3.07}$ | $54.73_{\pm3.99}$ | $50.25_{\pm5.57}$ | $48.83_{\pm5.95}$ |

Table 15: Performance of different attack methods under varying budgets on `Pubmed` dataset.

| Attack | 0.01 | 0.02 | 0.05 | 0.10 | 0.15 |
|---|---|---|---|---|---|
| DICE | 79.00 | 78.76 | 78.69 | 78.06 | 77.90 |
| PGA | 74.61 | 70.20 | 58.44 | 48.18 | 47.37 |
| GRBCD | 76.14 | 73.56 | 64.51 | 54.63 | 49.37 |
| PRBCD | 74.99 | 71.90 | 64.16 | 55.54 | 49.32 |
| EvA | **72.60** | **68.35** | **56.15** | **42.93** | **40.46** |

Table 18: Comparison of classification accuracy (%) on the `CoraML` dataset under EvA and (Dai et al., 2018b) across varying perturbation budgets $\epsilon$.

| Attack | Clean | 0.01 | 0.02 | 0.05 | 0.1 | 0.15 | 0.2 |
|---|---|---|---|---|---|---|---|
| (Dai et al., 2018b) | $81.07_{\pm2.07}$ | $78.50_{\pm1.66}$ | $76.66_{\pm2.22}$ | $72.53_{\pm1.91}$ | $68.75_{\pm1.45}$ | $65.34_{\pm1.20}$ | $63.27_{\pm2.47}$ |
| EvA | $81.07_{\pm2.07}$ | $\mathbf{74.80_{\pm1.50}}$ | $\mathbf{68.97_{\pm1.58}}$ | $\mathbf{52.95_{\pm1.91}}$ | $\mathbf{41.99_{\pm2.06}}$ | $\mathbf{37.65_{\pm2.74}}$ | $\mathbf{35.37_{\pm2.38}}$ |

the perturbation by referencing only the upper triangular indexing.

$$r = n - 2 - \left\lfloor \frac{\sqrt{-8l + 4n(n-1) - 7}}{2} - 0.5 \right\rfloor$$

$$c = 1 + l + r - \frac{n(n-1)}{2} + \left\lfloor \frac{(n-r)(n-r-1)}{2} \right\rfloor \qquad (2)$$

The advantage of this solution is that it can also be implemented in a vectorized way, making everything parallelizable.

**Attacking robustness certificates.** We define a randomized model as a convolution of the original model and a smoothing scheme. Namely the procedure or smooth inference is to add a noise (defined by the smoothing scheme) to the input, and evalute the model on the noisy input. The output of the smooth classifier is the probability of the top class over realizations of the noise (this is the output of the smooth classifier binary certificate; for confidence certificate the output is the expected softmax scores). The smoothing scheme $\xi : \mathcal{X} \mapsto \mathcal{X}$ is a randomized function mapping the given input to a random nearby point. For graph structure, we use the sparse smoothing certificate (Bojchevski et al., 2020), which certifies whether within $\mathcal{B}_{r_a, r_d}$ the prediction of the smooth model remains the same. Here $r_a$ is the maximum number of possible additions, and $r_d$ is the maximum number of edge deletions. The smoothing function is defined by two Bernoulli parameters $p_+$, and $p_-$; i.e. for each entity of $\boldsymbol{A}$, if it is zero, it will be toggled with $p_+$ probability and otherwise with $p_-$. The same smoothing scheme (and threat model) can be defined for features if the feature space is also binary and sparse. Setting $p_+ = p_-$ reduces the certificate to uniform smoothing certificate for $\ell_1$ ball.

Smoothing certificates require black-box access to the model $f$. As described above the smooth classifier is defined as $\bar{f}_y(\boldsymbol{x}) = \mathbb{E}[\mathbb{I}[f(\xi(\boldsymbol{x})) = y]]$ - each random sample $\boldsymbol{x}'$ is one vote for class $f(\boldsymbol{x}')$ and $\bar{f}_y$ is the proportion of votes for class $y$. Regardless of the model $f$, the smooth model $\bar{f}$ changes slowly around the input. Let $p = \bar{f}_y(\boldsymbol{x})$; for the smooth classifier we can find a lower bound probability $\underline{p} \geq \min_{\tilde{\boldsymbol{x}} \in \mathcal{B}(\boldsymbol{x})} \bar{f}_y(\tilde{\boldsymbol{x}})$ and define the certificate as a decision function $\mathbb{I}[\underline{y} \geq 0.5]$. This decision function ensures that the predicted class still remains the top-class for any point within the threat model. For details including the optimization function and how to compute certified lower bound see (Bojchevski et al., 2020).

Whether a node (an input in general) is certified reduces to whether the smooth prediction probability for the input is above a threshold $\bar{p}$. This is due to the non-decreasing property of the certified ratio with respect to $\bar{p}$. Additionally since the certificate is only a function of the probability and not the input, we can find this value easily via binary search. Therefore our objective is to decrease the probability of the smooth classifier below $\bar{p}$ for as many node as possible.

**Adaptive sampling for certificate attack.** Statistical rigor is not a necessity while attacking the certificate. Therefore, we can reduce the sampling rate to a low number while finding the perturbation. Later to ensure that our attack has reduces the certified ratio we again follow the proper certificate configuration. During the attack, we can reduce the cost of resampling by only resampling the subset of the graph that was perturbed. In other words, we initialize the search by computing and storing samples $\boldsymbol{A}_1, \ldots, \boldsymbol{A}_m$ from the clean graph, and for each perturbation $\tilde{\boldsymbol{A}}$ we only need to resample the edges in $\boldsymbol{A} \triangle \tilde{\boldsymbol{A}}$. Specifically for any edge removed from the graph during perturbation we update original sample $\boldsymbol{A}_1, \ldots, \boldsymbol{A}_m$ with $p_+$ Bernoulli samples in the same index of the added edge. Similar process is done with $p_-$ random edge removals for edges added in the perturbation. We substitute those samples in the same entry of $\boldsymbol{A}_1, \ldots, \boldsymbol{A}_m$, and by running this process $|\delta|$ times, we assume that $\tilde{\boldsymbol{A}}_1, \cdots \tilde{\boldsymbol{A}}_m$ are representative as a new set of $m$ samples for $\tilde{\boldsymbol{A}}$. This adaptive sampling reduces the number of random computations from $m \cdot n^2$ to $m \cdot |\delta|$, which is significantly lower. Surely, to evaluate the final perturbation (the reported effectiveness), we don't use this approach, as it is statistically flawed and only applicable to reduce the computation during the attack.

## D.1   ALGORITHM FOR EVA

Here we provide the algorithm and the pseudo-code for EvA. The input to our algorithm is the set of following variables: Graph $G = (\boldsymbol{X}, \boldsymbol{A})$ with $|\mathcal{V}| = n$, trained model $f$, attacked node set $\mathcal{V}_{\text{att}}$, labels $y_{\text{att}}$, global budget $\epsilon$. We set the followings as our hyper parameters: the fitness function $\text{fit}(\cdot)$, population size $n_p$, number of iterations $n_T$, tournament size $n_{\text{tour}}$, mutation rate $p_{\text{mut}}$, number of crossover joints $k_{\text{cross}}$.

Our population at each iteration is a matrix $\boldsymbol{S} \in [n]^{\delta \times n_p}$ where $n_p$ is the population size. We define $\mathcal{I}_{\text{init}}$ as the set of all possible edges that have one endpoint in $\mathcal{V}_{\text{att}}$. This set is used to form the initial matrix $\boldsymbol{S}$ due to an observation showing that initializing from these edges considerably improves the initial performance.

Before introducing the algorithm we define the following necessary blocks: (i) the function that converts a population element to a perturbation over the graph.  (ii) The tournament function that takes random subsets from the existing population and pick the top element from them. This function specifically chooses which elements in the population are remaining for the next iteration. And, (iii) the crossover function that designs a new element based on two existing elements in the population. All three functions are defined in Algorithm 1.

---

**Algorithm 1** Helper functions for EvA

---

**Ensure:** Adversarial adjacency $\tilde{A}_{\text{att}}$
1: Bijection $\Pi : \{(u,v) : 1 \le u < v \le n\} \to \{1, \ldots, n(n-1)/2\}$ and $\Pi^{-1}$     ▷ Index map

2: **function** APPLYPERTURBATION($A, s$)
3:     Initialize sparse matrix $P_s \in \{0,1\}^{n \times n}$ as all zeros
4:     **for** $t = 1, \ldots, \delta$ **do**
5:         $(i,j) \leftarrow \Pi^{-1}(s[t])$ with $i < j$
6:         $P_s[i,j] \leftarrow P_s[i,j] \oplus 1;$    $P_s[j,i] \leftarrow P_s[i,j]$
7:     **end for**
8:     **return** $\tilde{A} \leftarrow A \oplus P_s$             ▷ Edge flips are applied via XOR
9: **end function**

10: **function** TOURNAMENTSELECT($S, n_{\text{tour}}$)
11:     Sample a multiset $\mathcal{C}$ of $n_{\text{tour}}$ candidates from $S$ uniformly at random
12:     Return the fittest candidates in $\mathcal{C}$: $\arg\min_{s \in \mathcal{C}} \text{acc}(f(X, \text{ApplyPerturbation}(A, s)))$
13: **end function**

14: **function** CROSSOVER($s^{(1)}, s^{(2)}, k_{\text{cross}}$)
15:     Let $j_1, \ldots, j_{k_{\text{cross}}} := \text{RandomChoice}([\delta - 1])$ and sorted increasingly.
16:     Initialize $s_{\text{child},1} \leftarrow s^{(1)}, s_{\text{child},2} \leftarrow s^{(2)}$
17:     parent $\leftarrow 1$
18:     $j_0 \leftarrow 0, j_{k_{\text{cross}}+1} \leftarrow \delta$
19:     **for** $m = 1, \ldots, k_{\text{cross}}$ **do**
20:         parent $\leftarrow 3 - $ parent         ▷ Alternate between 1 and 2
21:         **for** $t = j_{m-1} + 1, \ldots, j_m$ **do**
22:             $s_{\text{child},1}[t] \leftarrow s^{(\text{parent})}[t]$
23:             $s_{\text{child},2}[t] \leftarrow s^{(3 \text{ - parent})}[t]$
24:         **end for**
25:     **end for**
26:     **return** $s_{\text{child},1}, s_{\text{child},1}$
27: **end function**

---

In the definition of the crossover function in Algorithm 1 there are two parents and two child elements. To flip the selected parent we use $3 - \text{parent}$ which changes from the first parent to the second and vice versa.

Other than the above functions, GA uses another function called mutation. This function applies random changes to the population enabling more exploration in the space of solutions. The following is a naive mutation function that is also used as our baseline equivalent to Dai et al. (2018a).

This mutation function is very less effective compared to the adaptive targeted mutation introduced in § 3. This is since the naive mutation proposes perturbations that are outside of the receptive field of $\mathcal{V}_{\text{att}}$ which does not introduce any change in the accuracy specially for low $\epsilon$. Here we introduce our adaptive targeted mutation which follows two modifications (i) any edge we add during mutation must have one endpoint in $\mathcal{V}_{\text{att}}$, and (ii) if a node in $\mathcal{V}_{\text{att}}$ is already misclassified there is no need to add edges connected to it anymore. The algorithm for our adaptive targeted mutation is in Algorithm 3.

With all the blocks defined in Algorithm 4 we provide the algorithm for EvA.

---

**Algorithm 2** Naive mutation function

---

1: **function** NAIVEMUTATION($s, p_{\mathrm{mut}}, \mathcal{E}$) $\triangleright$ $\mathcal{E}$ = set of all unordered node pairs $(u, v)$ with $u < v$
2:      **for** $t = 1, \ldots, \delta$ **do**
3:          **if** Bernoulli($p_{\mathrm{mut}}$) is 1 **then**
4:              Sample $(u, v) \in \mathcal{E}$ uniformly
5:              $s[t] \leftarrow \Pi(u, v)$                  $\triangleright$ Unbiased random mutation over all possible edges
6:          **end if**
7:      **end for**
8:      **return** $s$
9: **end function**

---

**Algorithm 3** Adaptive targeted mutation function

---

1: **function** ADAPTIVETARGETEDMUTATION($s, p_{\mathrm{mut}}, \mathcal{V}_{\mathrm{att}}, \mathcal{V}_{\mathrm{flipped}}$)
2:      $\mathcal{U} \leftarrow \mathcal{V}_{\mathrm{att}} \setminus \mathcal{V}_{\mathrm{flipped}}$              $\triangleright$ Attacked nodes whose label is not yet flipped
3:      **for** $t = 1, \ldots, \delta$ **do**
4:          **if** Bernoulli($p_{\mathrm{mut}}$) is 1 **then**
5:              Sample $u \in \mathcal{U}$ uniformly
6:              Sample $v \in \mathcal{V}$ uniformly
7:              $s[t] \leftarrow \Pi(\min(u, v), \max(u, v))$      $\triangleright$ Targeted mutation: at least one endpoint in $\mathcal{V}_{\mathrm{att}} \setminus \mathcal{V}_{\mathrm{flipped}}$
8:          **end if**
9:      **end for**
10:      **return** $s$
11: **end function**

---

### D.2 HOW SOLUTION EVOLVES

s We further conduct an ablation study on the solutions found by EvA and PRBCD under a specific budget of 10%. In this experiment, we keep all hyperparameters of EvA and PRBCD fixed and run them across 10 different seeds. We then compare the average solutions generated by each adversary. The left figure in Fig. 9 shows the number of connections across different labels. In both cases, the methods focus more on label 5 than on the others, but EvA distributes the connections more uniformly compared to PRBCD. The middle figure illustrates the nodes with original degrees ranging from 1 to greater than 8. The results indicate that, in both attacks, most of the budget is spent connecting to low-degree nodes. However, compared to PRBCD, EvA allocates more of the budget to higher-degree nodes. Additionally, we calculate the margin loss for each node in the original graph and discretize them into eight levels. As shown in the right figure of Fig. 9, EvA allocates more of the budget to higher-margin nodes, resulting in a non-trivial solution that achieves a better optimum. Finally, it seems that EvA identifies solutions that differ from greedy-based heuristic, which usually only targets low-degree or low-margin nodes.

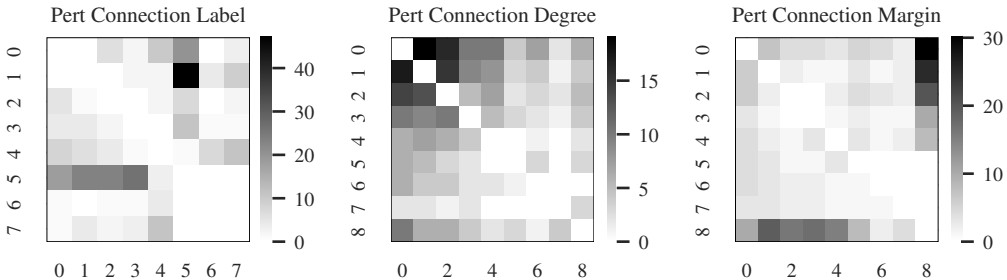

Figure 9: The upper triangle of each heatmap represents the perturbation connections for PRBCD, the lower triangle corresponds to the same for EvA, and the diagonal is set to zero.

---

**Algorithm 4** EvA: Evolutionary Attack on Graphs

---

1: $\mathcal{I}_{\text{init}} \leftarrow \mathcal{E}[\mathcal{V}_{\text{att}} : \mathcal{V}]$         ▷ Search space: Edges touching attacked nodes
2: $\delta \leftarrow \lfloor \varepsilon \cdot |\mathcal{I}_{\text{init}}| \rfloor$            ▷ Perturbation budget
3: $\boldsymbol{S} \leftarrow \text{InitializePopulation}(\mathcal{I}_{\text{init}}, P)$      ▷ Population of size $P$
4: $\tilde{\boldsymbol{A}}_{\text{best}} \leftarrow \boldsymbol{A}; \quad F_{\text{best}} \leftarrow -\infty$
5: **for** $t = 1, \ldots, T$ **do**
6:   $\mathcal{R}_{\boldsymbol{S}} \leftarrow \emptyset$
7:   **for** $\boldsymbol{s}_i \in \boldsymbol{S}$ **do**
8:    $\tilde{\boldsymbol{A}}_i \leftarrow \text{ApplyPerturbation}(\boldsymbol{A}, \boldsymbol{s}_i)$
9:    $F_i \leftarrow \text{fit}(f(\boldsymbol{X}, \tilde{\boldsymbol{A}}_i), \mathcal{V}_{\text{att}})$
10:    $\mathcal{R}_{\boldsymbol{S}} \leftarrow \mathcal{R}_{\boldsymbol{S}} \cup \{F_i\}$
11:    **if** $F_i > F_{\text{best}}$ **then**
12:     $F_{\text{best}} \leftarrow F_i; \quad \tilde{\boldsymbol{A}}_{\text{best}} \leftarrow \tilde{\boldsymbol{A}}_i$
13:    **end if**
14:   **end for**
15:   $\boldsymbol{S}_{\text{new}} \leftarrow \{\arg\max_{\boldsymbol{s} \in \boldsymbol{S}} F(\boldsymbol{s})\}$
16:   **while** $|\boldsymbol{S}_{\text{new}}| < P$ **do**
17:    $(\boldsymbol{s}^{(1)}, \boldsymbol{s}^{(2)}) \leftarrow \text{TournamentSelect}(\boldsymbol{S}, \mathcal{R}_{\boldsymbol{S}}, n_{\text{tour}})$
18:    $\boldsymbol{s}_{\text{c1}}, \boldsymbol{s}_{\text{c2}} \leftarrow \text{Crossover}(\boldsymbol{s}^{(1)}, \boldsymbol{s}^{(2)}, k_{\text{cross}})$
19:    **for** $\boldsymbol{s}_{\text{child}} \in \{\boldsymbol{s}_{\text{c1}}, \boldsymbol{s}_{\text{c2}}\}$ **do**
20:     $\tilde{\boldsymbol{A}}_{\text{child}} \leftarrow \text{ApplyPerturbation}(\boldsymbol{A}, \boldsymbol{s}_{\text{child}})$
21:     $\hat{\boldsymbol{y}} \leftarrow f(\boldsymbol{X}, \tilde{\boldsymbol{A}}_{\text{child}})$         ▷ Forward pass
22:     $\mathcal{V}_{\text{flipped}} \leftarrow \{v \in \mathcal{V}_{\text{att}} : \hat{y}_v \neq y_v\}$
23:     $\boldsymbol{s}_{\text{child}} \leftarrow \text{AdaptiveTargetedMutation}(\boldsymbol{s}_{\text{child}}, p_{\text{mut}}, \mathcal{V}_{\text{flipped}})$
24:     $\boldsymbol{S}_{\text{new}} \leftarrow \boldsymbol{S}_{\text{new}} \cup \{\boldsymbol{s}_{\text{child}}\}$
25:    **end for**
26:   **end while**
27:   $\boldsymbol{S} \leftarrow \boldsymbol{S}_{\text{new}}$
28: **end for**
29: **return** $\tilde{\boldsymbol{A}}_{\text{adv}} \leftarrow \tilde{\boldsymbol{A}}_{\text{best}}$

---

We also track the perturbation connections at different steps of EvA to see how the best solution evolves during optimization. In Fig. 10, as you can see, at the beginning it connects nodes with high margins to those with low margins which suggest it start with simply affecting the side of the node which is more vulnerable (lower margin), but as it proceeds, it also optimizes connections between high-margin and medium-margin nodes. This suggests that it attempts to degrade the performance of nodes from both sides of the edge increasing the efficiency of the budget usage.

In Fig. 11, we first observe that EvA spends most of its budget on low-degree nodes (which are easier to attack). We further observe how the method gradually reduces its focus on higher-degree nodes (which are harder to attack), while still targeting some high-degree nodes that remain vulnerable. Finally, in Fig. 12, we track the evolution of label focus: label 4 (which corresponds to label 5 in the previous figure) receives more attention throughout all the steps. In the beginning there are lots of within-class perturbations which are gradually reduced during the evolution.

In Fig. 13 [Left, Middle], we measure the number of unique edges and nodes across the entire population (all 512 members) at different steps. As expected, the number of unique edges and node are decreasing. This indicates that the attack is converging. We also measured the number of nodes and edges that are repeated in every member of the population (i.e., the size of the intersection across the entire population) in Fig. 13 [Right]. Increasing the number of nodes and edges shared across the whole population means the algorithm is converging and moving toward exploitation, while having fewer shared nodes and edges indicates exploration. Fig. 13 [Right] reveals an interesting pattern: during the first 500 iterations, EvA alternates between exploration and exploitation, repeatedly introducing new edges and then reverting. After around iteration 500, exploration decreases and the algorithm shifts more toward exploitation.

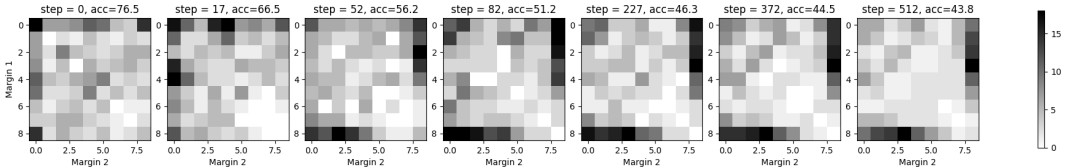

Figure 10: Visualization of how perturbation connections evolve throughout the optimization steps of EvA. Early in training, connections primarily link high-margin nodes [buckets 7-8] to low-margin nodes [buckets 1-2]. As optimization progresses, the method increasingly forms connections between high- and medium-margin nodes.

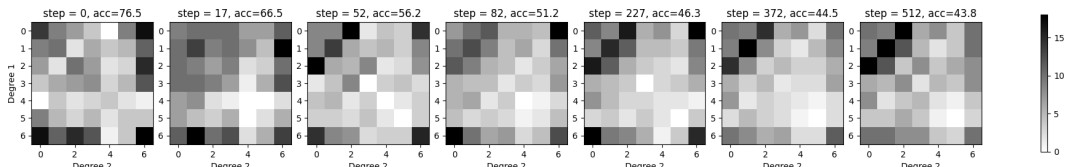

Figure 11: The gradual evolution of the perturbation and the distribution of perturbation connections with respect to node degree.

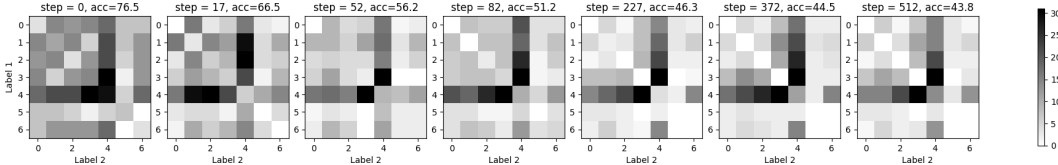

Figure 12: Visualization of how perturbation connections shift over optimization steps, showing how the method progressively redistributes its focus among different label pairs.

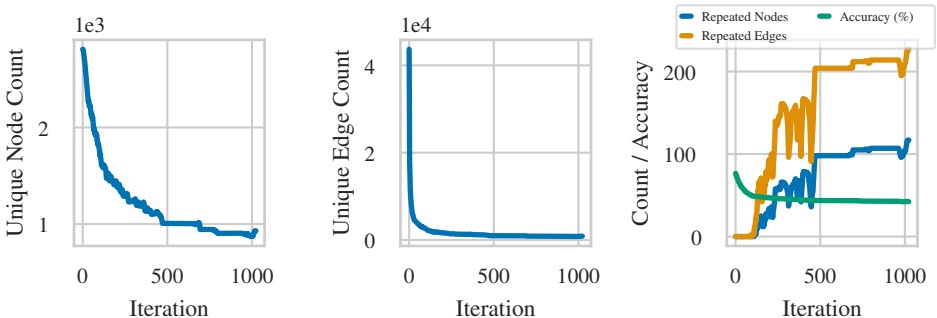

Figure 13: [Left] shows the number of unique nodes across the population, [Middle] shows the number of unique edges across all 512 individuals. [Right] shows the edges and nodes that appear in every individual.

In Fig. 14, we present the perturbation graph that is the concatenation of all perturbations across the entire population. The position of each node is based on a 2D representation of the logits, obtained using the t-SNE algorithm for dimensionality reduction. The edge widths indicate how many times an edge is repeated in the population, the node size represents the number of perturbations that include this node, the node colors correspond to different labels, and square nodes denote training nodes, while circular nodes denote attack nodes.

As shown, at the beginning of exploration all of the edges have low width. Toward the end, the edges become thicker and some nodes grow larger. Interestingly, our attack exploits nodes whose latent representation is located in the "wrong" cluster, perturbing these nodes is likely more effective. Finally, we observe more connections originating from the pink-labeled nodes which are more dispersed in the embedding space. Intuitively, this allows EvA to gain leverage in regions where the representation is loosely organized.

Finally, we look at the solutions that PRBCD and EvA find, and we compare how many nodes are directly influenced on average over 10 different runs. This evaluation revealed that EvA affects approximately 282 nodes on average, whereas PRBCD influences only about 208 nodes directly in the final attack solution. This suggests that PRBCD still overspends part of its budget on a small number of nodes.

We further provide Table 19 and Table 20, which shows the degree of perturbation for each node, i.e. $\sum_j \boldsymbol{P}_{ij}$ for all $i$, over 10 runs for both EvA and PRBCD. For example, in Table 19, we see that in the first run we have 251 nodes with only a single perturbation, 27 nodes with 2 perturbations, 5 nodes with 3 perturbations, and so on. These results indicate that EvA attempts to change nodes more efficiently usually modifying them with just one edge flipping. It uses less two-flips and three-flips compared to PRBCD (Table 20), and only rarely performs four or five flips on a single node. PRBCD, on the other hand, almost always overspends its budget on one or two nodes with very high degrees of modification. This reduces the opportunity to distribute its budget across more nodes, thereby lowering the number of nodes that PRBCD directly influences.

Table 19: The counts of perturbation degrees over 10 runs of EvA on `CoraML`.

| Run | 1.0 | 2.0 | 3.0 | 4.0 | 5.0 |
|---|---|---|---|---|---|
| 1 | 251 | 27 | 5 | 1.0 | 0.0 |
| 2 | 246 | 27 | 8 | 0.0 | 0.0 |
| 3 | 242 | 38 | 2 | 0.0 | 0.0 |
| 4 | 250 | 29 | 4 | 1.0 | 0.0 |
| 5 | 244 | 34 | 4 | 0.0 | 0.0 |
| 6 | 259 | 25 | 2 | 1.0 | 1.0 |
| 7 | 246 | 33 | 4 | 0.0 | 0.0 |
| 8 | 241 | 29 | 7 | 1.0 | 0.0 |
| 9 | 251 | 30 | 3 | 1.0 | 0.0 |
| 10 | 229 | 43 | 3 | 0.0 | 0.0 |

Table 20: The counts of perturbation degrees over 10 runs of PRBCD on `CoraML`.

| Run | 1.0 | 2.0 | 3.0 | 4.0 | 5.0 | 6–10 | 11–30 |
|---|---|---|---|---|---|---|---|
| 1 | 140 | 47 | 13 | 8 | 0.0 | 0 | 1 |
| 2 | 137 | 37 | 19 | 4 | 1.0 | 0 | 1 |
| 3 | 136 | 50 | 17 | 7 | 1.0 | 0 | 0 |
| 4 | 145 | 48 | 12 | 5 | 1.0 | 2 | 0 |
| 5 | 149 | 35 | 15 | 6 | 4.0 | 5 | 0 |
| 6 | 149 | 44 | 13 | 3 | 1.0 | 3 | 0 |
| 7 | 154 | 36 | 13 | 2 | 3.0 | 5 | 0 |
| 8 | 127 | 46 | 17 | 4 | 1.0 | 2 | 0 |
| 9 | 163 | 33 | 11 | 5 | 2.0 | 2 | 1 |
| 10 | 130 | 54 | 17 | 3 | 3.0 | 3 | 0 |

## D.3 TIME ANALYSIS

We run an ablation study comparing PRBCD, and EvA for wall clock time and memory. In EvA, the number of steps controls the time and the size of the population (assuming all population is evaluated at once using stacked inference) controls the memory. Similarly for PRBCD, time is controlled by the number of epochs controls the time and memory is a function of block size. We evaluate EvA with different numbers of steps, population sizes, and parallel evaluations, and PRBCD with varying numbers of epochs and block sizes on the `Pubmed` dataset. Fig. 15 (left) shows the results for EvA and PRBCD in terms of memory usage, wall clock time and method performance. Our method demonstrates comparable performance within the same level of wall clock time (less than a minute). Moreover, by increasing the wall clock time and memory either by a larger population size or more steps, EvA enhances its performance. This is while PRBCD has an almost constant trend given more time or memory.

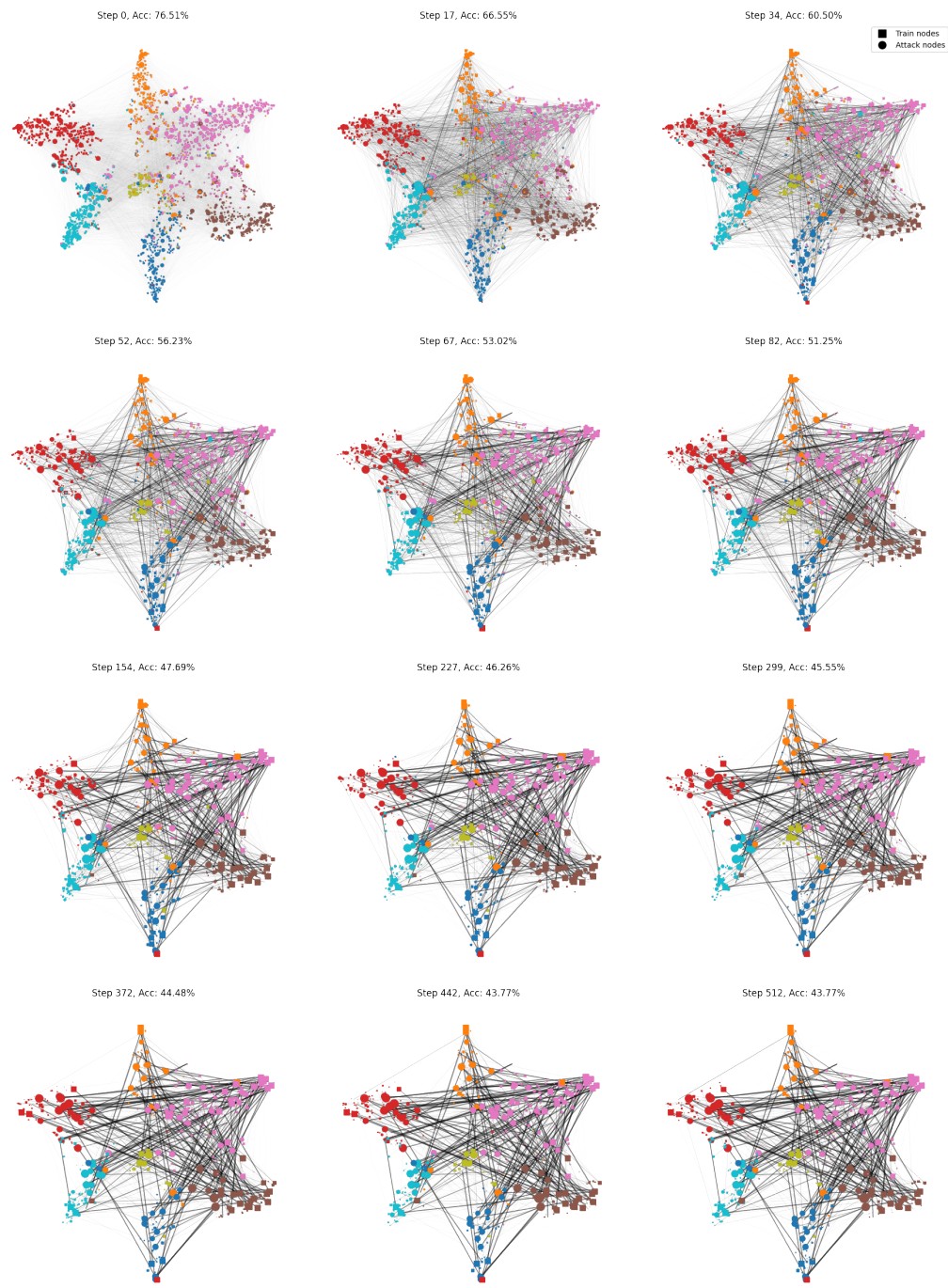

Figure 14: Population-wide perturbation graph showing t-SNE-projected node positions, with edge thickness reflecting repetition, node size indicating degree, and colors denoting labels; training nodes are squares and attack nodes are circles.

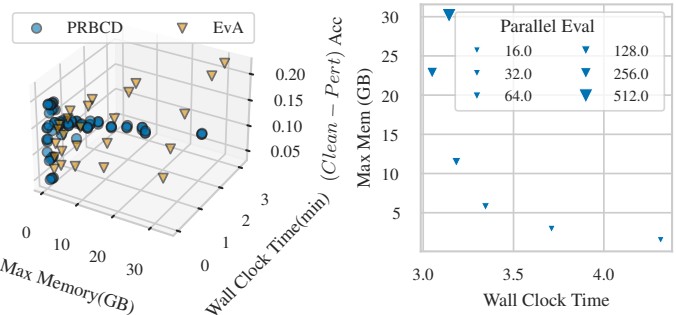

Figure 15: Comparing the memory usage between EvA and PRBCD.

Additionally, in Fig. 15, we highlight how our framework provides a trade-off between time and memory for achieving the same level of accuracy by varying the number of parallel evaluations. For each point in the figure, we observe roughly the same performance; however, the methods differ in memory usage due to different number of parallel evaluation, leading to variations in wall clock time.

### D.4 DIVIDE AND CONQUER

Computing gradients w.r.t. all elements in matrix $A$ is computationally intensive. And as the graph grows the elements for which we should compute and store gradient increases quadratically. As a remedy PRBCD applies a block-coordinate gradient descent where in each iteration the gradients are computed over a subset of indices in $A$. Intuitively PRBCD works under this assumption that a relaxation from $A$ to a (random) subset of adjacency matrix does change the optimal solution by a high margin. In Fig. 16 for `Ogbn-Arxiv` dataset, we increased the block size of PRBCD, from the suggested 3M to 10M, and still the result is far from EvA. Beyond that block size could not fit into the memory.

While EvA works with sparse representation of $A$ still the search space ($2^{|A|}$) grows exponentially with the number of nodes. Via divide and conquer (D&C) we apply relaxation where we assume a sequential search for optimal attack targeting disjoint subsets of $\mathcal{V}_{att}$ does not result far from the optimal solution for the entire $\mathcal{V}_{att}$. Therefore we divide this set into $k$ disjoint subsets and run the attack over each subset separately. Our attack applies on subsets in a sequential order meaning that after attacking one subset, the attacked graph is assumed to be the clean baseline to attack the other set. At the final round, the attack carries all the perturbations applied on $\mathcal{V}_{att}$. To ensure the validity of the result, we divide the budget according to the edges connected to each subset.

As our D&C approach is applicable regardless of the attack algorithm we can similarly apply it to PRBCD. Although it does not outperform EvA, we show that adding D&C to PRBCD increases the performance by a significant margin.

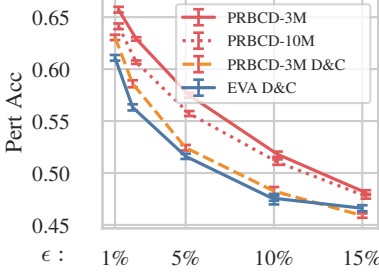

Figure 16: Effect of Divide and Conquer on EvA and PRBCD.

# E  DATASETS AND MODELS, AND HYPERPARAMETERS

## E.1  STATISTICS OF DATASETS

In our experiments, we mainly conduct experiments on the commonly used graph datasets: `CoraML`, `Citeseer`, and `Pubmed`, which are all representative academic citation networks. Their specific characteristics are as follows:

**CoraML.** The `CoraML` dataset contains 2,810 papers as nodes, with citation relationships between them as edges, resulting in 7,981 edges. Each paper is categorized into one of 7 classes corresponding to different subfields of machine learning. Each node is represented by a 1,433-dimensional bag-of-words (BoW) feature vector derived from the words in the titles and abstracts of the papers.

**Citeseer.** The `Citeseer` dataset is also an academic citation network dataset consisting of 3,312 papers from 6 subfields of computer science and a total of 4,732 citation edges. Similar to `CoraML`, each paper as a node is represented by a BoW feature vector with a dimensionality of 3,703.

**Pubmed.** The `Pubmed` dataset is derived from a citation network of biomedical literature that contains 19,717 papers as nodes and 44,338 citation edges. Each paper is categorized into one of 3 classes based on its topic. The node features in `Pubmed` are 500-dimensional vectors.

**Amazon-Computers and Amazon-Photo.** The `Amazon-Computers` and `Amazon-Photo` datasets consists of two networks of Amazon Computers and Amazon Photo. In these networks, nodes represent individual goods sold on Amazon, and edges indicate that two products are frequently purchased together. Each node is accompanied by bag-of-words features derived from product reviews, providing a textual representation of the item's description and customer feedback. The task is predicting the product category.

Table 21: Dataset statistics.

| Dataset | Nodes | Edges | Features | Classes |
|---|---|---|---|---|
| **CoraML** | 2,810 | 7,981 | 1,433 | 7 |
| **Citeseer** | 3,312 | 4,732 | 3,703 | 6 |
| **Pubmed** | 19,717 | 44,338 | 500 | 3 |
| **Amazon-Computers** | 13,752 | 491,722 | 767 | 10 |
| **Amazon-Photo** | 7,650 | 238,162 | 745 | 8 |

## E.2  DETAILS OF MODELS

In the following sections, we detail the hyperparameters and architectural details for the models performed in this paper.

**GCN.** We utilize a two-layer GCN with 64 hidden units. A dropout rate of 0.5 is applied during training.

**GAT.** Our GAT model consists of two layers with 64 hidden units and a single attention head. During training, we apply a dropout rate of 0.5 to the hidden units, but no dropout is applied to the neighborhood.

**APPNP.** We use a two-layer MLP with 64 hidden units to encode the node attributes. We then apply generalized graph diffusion, using a transition matrix and coefficients $\gamma_K = (1 - \alpha)K$ and $\gamma_l = \alpha(1 - \alpha)l$ for $l < K$.

**GPRGNN.** Similar to APPNP, we employ a two-layer MLP with 64 hidden units for the predictive part. We use the symmetric normalized adjacency matrix with self-loops as the transition matrix and randomly initialize the diffusion coefficients. We consider a total of $K = 10$ diffusion steps, with $\alpha$ set to 0.1. During training, we apply a dropout rate of 0.2 to the MLP, while no dropout is applied to the adjacency matrix. Unlike the method in Chien et al. (2021), we always learn the diffusion coefficients with weight decay, which acts as a regularization mechanism to prevent the coefficients from growing indefinitely.

Table 22: Hyper-parameters for PRBCD, LRBCD, and EvA.

| Hyper-parameter | PRBCD | LRBCD | Hyper-parameter | EvA |
|---|---|---|---|---|
| Epochs | 500 | 500 | No. Steps | 500 |
| Fine-tune Epochs | 100 | 0 | Mutation Rate | 0.01 |
| Keep Heuristic | WeightOnly | WeightOnly | Tournament Size | 2 |
| Search Space Size | 500,000 | 500,000 | Population Size | 1,024 |
| Loss Type | Tanh-Margin | tanh-Margin | No. Crossovers | 30 |
| Early Stopping | N/A | False | Mutation Method | Adaptive |

**SoftMedian GDC.** We follow the default configuration from Geisler et al. (2023), using a temperature of $T = 0.2$ or the SoftMedian aggregation, with 64 hidden dimensions and a dropout rate of 0.5. We fix the Personalized PageRank diffusion coefficient to $\alpha = 0.15$ and apply a top $k = 64$ sparsification. During the attacks, the model remains fully differentiable, except for the sparsification of the propagation matrix.

**MLP.** We design the MLP following the prediction module of GPRGNN and APPNP, incorporating two layers with 64 hidden units. During training, we apply a dropout rate of 0.2 to the hidden layer.

### E.3 HYPERPARAMETER SETUP

In EvA we set the capacity of the computation to the same as the population, this means that all perturbations within a population are in one combined inference. However, in some cases where the graph is large (e.g. Pubmed), we reduce this number.

Table 22 shows the hyper-parameter selection in almost all experiments. We only change the population number in some experiments, like certificate attacks, to reduce the computation. E.g., in the certificate attack, the population is reduced by a factor of 10. Finally, all of the experiments has been run on one NVIDIA H100 GPU.

### E.4 ATTACK HYPERPARAMETERS

To assess the robustness of GNNs, we utilize the following attacks and hyperparameters. Based on Geisler et al. (2023), we also select the Tanh-Margin loss as the attack objective.

**PRBCD.** We closely adhere to the setup outlined by Geisler et al. (2023). A block size of 500,000 is used with 500 training epochs. Afterward, the model state from the best epoch is restored, followed by 100 additional epochs with a decaying learning rate and no block resampling. Additionally, the learning rate is scaled according to $\delta$ and the block size, as recommended by Geisler et al. (2023).

**LRBCD.** The same block size of 500,000 is used with 500 training epochs. The learning rate is scaled based on $\delta$ and the block size, following the same approach as PRBCD. The local budget is consistently set as 0.5.

**EvA.** We set the population size to 1024 in most cases. Our mutation rate is 0.01, and increasing this number breaks the balance between exploration and exploitation, leading to less effective attacks. We run each attack for 500 iterations in most cases. In cases like certificate attacks, which are time-consuming, we reduce this number to 100. The details are summarized in Table 22.

For Ogbn-Arxiv dataset we divide the $\mathcal{V}_{\text{att}}$ to $k_{\text{dc}} = 98$ subsets where each division includes 500 vertices. There we set the population size to 45 candidates. The $\delta_i$ for subset $i$ is set to $\epsilon_i = \epsilon \cdot |\mathcal{E}[\mathcal{V}_i : \mathcal{V}]|$. We also reduce the number of iterations for EvA to 300 for each subset while for PRBCD this number remains 500.

**EvA-Local.** All hyper-parameters are the same as EvA. An additional hyper parameter is $t_{\text{warm}}$ which is the number of initial steps where the random local projection is applied instead of the frequency score-based local projection. This number is set to 50 for Pubmed dataset. Interestingly even without this projection, EvA-Local outperforms LRBCD for Citeseer and CoraML datasets. There we only remove random matchings from the nodes with degree violation until the total violation reaches 0. This approach does not work on the Pubmed dataset. The intuition is that since Pubmed

larger and denser compared to other datasets, for each candidate there are a lot of edges that have at least one endpoint violating the local constraint. Therefore removing many edges from a candidate, and replacing them by random edge adds a large random noise to each candidate at each iteration. Therefore the search is done over a very noisy setup.

**PGA.** For the PGA, we adopt the same setting as in Zhu et al. (2023). We use GCN as the surrogate model and tanhMarginMCE-0.5 as the loss type. The attack is configured with 1 greedy step, a pre-selection ratio of 0.1, and a selection ratio of 0.6. Additionally, the influence ratio is set to 0.8, with the selection policy based on node degree and margin.

