# OpenReview forum: "EvA: Evolutionary Attacks on Graphs"
_ICLR.cc/2026/Conference — ICLR 2026 Poster_

### Official Review · Reviewer_KDQh · 2025-10-31

**Soundness:** 3
**Presentation:** 3
**Contribution:** 3
**Rating:** 8
**Confidence:** 3

**Summary:**

The paper proposes EvA (Evolutionary Attack), a black-box, gradient-free attack on graph neural networks (GNNs) that directly solves the discrete edge-perturbation problem using a tailored genetic algorithm. Unlike dominant gradient-based attacks (PRBCD, LRBCD), EvA does not relax the adjacency matrix to a continuous space and can therefore optimize non-differentiable objectives such as accuracy, certified ratio, and conformal coverage.

**Strengths:**

The paper convincingly argues that gradient-based structure attacks are fundamentally misaligned with the discrete problem, e.g. gradients are local, ignore edge interactions, require relaxations, and can be obfuscated. The paper's empirical evaluation is comprehensive and the results are significant, not just marginal. EvA drastically outperforms all baselines, including the SOTA PRBCD, across multiple datasets.

**Weaknesses:**

The most significant weakness, which the authors acknowledge in the limitations, is the high query complexity. Genetic algorithms are inherently query-intensive.

**Questions:**

Could the authors provide a direct comparison of the total number of forward passes (queries) used by EvA versus the number of forward/backward passes used by PRBCD to achieve the results in Table 1?

---

> ### Author Response · Authors · 2025-11-25
>
> ***Q1: EvA versus the number of forward/backward passes used by PRBCD to achieve the results in Table 1?***
>
> We thank the reviewer for their thoughtful review. We would like to draw attention to **Figure 3 (right)**, where we report the number of forward passes used by **EVA** compared to the number of forward/backward passes required by **PRBCD** under the same memory constraints for the CoraML and PubMed datasets. As shown in the figure, EVA begins to outperform PRBCD after approximately **100 steps on CoraML** and **400 steps on PubMed**, and continues to surpass PRBCD beyond those points.
>
> For Table 1, this is not a memory equal experiment, but with a population size of 1024, we can process the entire population in just two forward passes. This results in a total of 500steps * 2 = 1000 forward. In contrast, PRBCD has been run for 500 steps, in which each step contains one forward and one backward, which makes it 500 * 2 = 1000 forward and backward in total.
>
> We would like to again thank the reviewer for reading our paper. We are happy to respond to any additional questions and feedback.

---

### Official Review · Reviewer_oVZ5 · 2025-11-01

**Soundness:** 2
**Presentation:** 1
**Contribution:** 2
**Rating:** 2
**Confidence:** 4

**Summary:**

This paper proposed a genetic adversarial attack. Experiments compared with a few gradient based attack are conducted with many ablation study.

**Strengths:**

**1.** Experiments in various aspects are done to validate the method performance with abundant figures for illustrations.

**Weaknesses:**

**1.** The presentation of the paper is bad. There's neither formulation nor algorithm written, or any figure to completely show the pipeline of the proposed attack. The description of method is just split around all section 3 and 4 without a clear introducing logic, instead just describing sentences and paragraphs concatenated. The methodology description highly relies on comparison with a previous baseline "PRBCD" which was not formulated introduced as well, making the part harder to follow. There are also too many verbal definitions which are lack of clear expression and only used once. In all, the bad writing makes me really hard to have a full view of the proposed method in details, and I recommend the author to rewrite and reformulate the paper entirely.

**2.** The experiments lack fully comparison with other works. The paper only compares with a few gradient based attack baselines, excluding experimental comparison with all other kind of attack by just stating "gradients method are SOTA" "out perform others". Considering the gradient baselines raised contain only one in 2023 while all others are before 2020, and the proposed method itself is indeed one of "beaten" genetic method, this lack of new baselines and ones from other attack types are unacceptable.

**Questions:**

Please see weakness.

---

> ### Author Response · Authors · 2025-11-25
>
> **W1:**
>
> We respectfully disagree with the reviewer's opinion. As the reviewer noted, we describe the methodology in Sections 3 and 4. In Section 3, we provide a comprehensive explanation of our proposed method, and in Section 4, we detail how our framework can be adapted to more complex settings, including two cases with entirely new objectives (certified robustness attack and conformal set size attack). To improve clarity, we have also added pseudocode for our algorithm in the revised version of the paper in Appendix D.1.
> Regarding the formulation of PRBCD, we believe it is not standard practice to provide a full description of a prior method. Instead, we explain the threat model in Section 2 and refer readers to the PRBCD paper for the complete details.
>
> We would also appreciate it if the reviewer could specify which parts they consider “verbal definitions.” Such clarification would help us update our paper in our revision.
>
> ---
>
> **W2: lack fully comparison with other works:**
>
> We refer the reviewer to Table 1 in the main paper and Table 15 in the appendix, where we report results comparing **EVA** with **DICE**, **PGA**, **PGD**, **PRBCD**, and **GRBCD**. In most experiments, we focus on PRBCD because it remains state-of-the-art and is one of the most stable attacks across different datasets and models.
>
> Second, we would like to draw the reviewer’s attention to **Table 4**, where we start from the method of Dai et al, the previous evolutionary search method, which has been surpassed by PRBCD, and incrementally add our contributions. This ablation demonstrates that evolutionary attacks can indeed be effective in practice, and that with our additional components, EVA outperforms PRBCD significantly.
>
> Finally, we would appreciate it if the reviewer could point us to specific papers or attack methods they believe should be included for comparison with EVA. This would help us strengthen the experimental section in future revisions.
>
> We would like to again thank the reviewer for reading our paper. We believe our responses address the raised concerns, and we hope this will be reflected in the updated assessment. We are happy to respond to any additional questions.

---

### Official Review · Reviewer_4Bnj · 2025-11-01

**Soundness:** 2
**Presentation:** 2
**Contribution:** 2
**Rating:** 4
**Confidence:** 4

**Summary:**

This paper introduces EvA (Evolutionary Attack), a new framework for edge-based adversarial attacks on GNNs through a discrete evolutionary search rather than gradient-based optimization. EvA formulates the attack as a genetic algorithm that evolves a population of perturbation candidates, where each candidate encodes a small set of edge flips. The algorithm iteratively applies Selection, Crossover, and Mutation. They also design a divide-and-conquer strategy to handle large graphs. Because it only requires model evaluations, not gradients, EvA is model-agnostic and applicable to black-box settings. EvA consistently achieves larger drops in classification accuracy than gradient-based attacks. The evolutionary framework can also attack non-differentiable objectives, where gradient-based methods cannot operate.

**Strengths:**

This paper uses a discrete evolutionary search method for graph edge-based adversarial attacks on GNNs, without the need of gradients, making it model-agnostic and applicable to black-box settings. The evolutionary framework can also attack non-differentiable objectives. They show strong empirical performance comparing with gradient-based methods.

**Weaknesses:**

1. The proposed evolutionary search is highly heuristic and not guaranteed to find globally optimal perturbations. Many of its design--mutation rate, crossover scheme, and selection strategy--lack principled justification or ablation analysis. It remains unclear which components are critical for performance and how sensitive the attack is to hyperparameter choices.
2.  The algorithm is difficult to follow from the current text presentation. Including clear pseudo-code or an algorithm box would greatly improve readability and reproducibility.
3. The scalability and efficiency analysis are underdeveloped (e.g., runtime, total queries, memory usage). Currently there's not statistics showing the time and memory consumptions of PRBCD and EvA on various datasets.
4. Using evolutionary search is not entirely new. The main novelty here lies in engineering and scaling rather than in conceptual advances. The paper would benefit from a clearer discussion of how its evolutionary design specifically differs from prior heuristic or search-based attacks.

**Questions:**

I wonder how EvA performs on larger graphs like arXiv, Products, and Papers 100M [1].

[1] W. Hu, M. Fey, M. Zitnik, Y. Dong, H. Ren, B. Liu, M. Catasta, and J. Leskovec. Open Graph Benchmark: Datasets for Machine Learning on Graphs. 2020.

---

> ### Author Response · Authors · 2025-11-25
>
> We thank the reviewer for taking the time to read our paper and for the review. We have outlined our responses below and hope they resolve your concerns.
>
> **W1: lack principled justification or ablation analysis. It remains unclear which components are critical for performance and how sensitive the attack is to hyperparameter choices.:**
>
> We thank the reviewer for taking the time to read our paper and for your review.
>
> We would like to draw the reviewer’s attention to the fact that we never claim our method finds *optimal* perturbations. Our only claim is that “current gradient-based attacks are still very far from optimal since EvA outperforms them by a notable margin,” a statement we support with empirical results demonstrating that EvA can indeed discover stronger perturbations.
>
> Furthermore, we provide ablation studies for every major design choice. First, we refer the reviewer to Table 4, where we gradually add each component of the attack mutation, modified fitness, sparse representation, and the divide-and-conquer strategy, and show how each contributes to the final performance. We also include an ablation on population size in Figure 3, illustrating how increasing the population affects performance. In Figure 2 (left and middle), we separately present the effects of mutation selection and the fitness function.
>
> We believe the nature of search implies that, given infinite computation, it will eventually find the correct solution. The point of our contribution is that we can find significantly better attacks *within practical memory and time budgets*, showing that our method is broadly useful. Additionally, all hyperparameters, such as the mutation rate or crossover rate, are kept fixed across all datasets and models (both defended and undefended), which further demonstrates that our method transfers well across different models and datasets.
>
> ----
>
> **W2: Including clear pseudo-code or an algorithm box would greatly improve readability and reproducibility.**
>
> We provide pseudo-code for our EvA as well as our adaptive targeted mutation (Check Appendix D.1) in the updated manuscript.
>
> ----
>
>
> **W3: statistics showing the time and memory consumptions of PRBCD and EvA on various datasets.**
>
> We would like to draw the reviewer’s attention to Figure 3 (right). In this figure, we compare the number of forward passes of EVA with the number of forward **and** backward passes of PRBCD on two datasets (CoraML and PubMed) under the same memory budget, in order to evaluate the query efficiency of our method. As the figure indicates, for smaller datasets such as CoraML, we are able to match PRBCD’s performance with fewer than 100 forward passes, and subsequently outperform it. As the graph size increases, the task naturally becomes more challenging. Accordingly, on PubMed, we begin to surpass PRBCD after roughly 400 forward passes and continue improving beyond that point. Additionally, we include a comparison of memory and time complexity in Figure 10 (Appendix) for the PubMed dataset, which provides further ablation analysis.
>
> We hope we address the concerns of the reviewer. We would be grateful if the reviewer could clarify more specifically what type of comparison they would like to see.
>
> -----
>
> **W4: The paper would benefit from a clearer discussion of how its evolutionary design specifically differs from prior heuristic or search-based attacks.**
>
> We thank the reviewer for raising this point. We fully agree that evolutionary search itself is not new, and we do not claim to “reinvent” evolutionary algorithms. In the same way that PRBCD is largely an engineering and scaling contribution on top of gradient-based attacks, our work focuses on engineering and scaling evolutionary search so that it becomes a ***practical*** and ***competitive* edge-based attack on graphs**.
>
> Our novelty lies in designing several modules that make this approach effective and usable in practice: the tailored mutation operators, the modified fitness function, the sparse perturbation representation, and the divide-and-conquer strategy for large graphs (which we also show can be used independently, also for PRBCD to improve its result). Table 4 provides a step-by-step ablation where we progressively add these components and show how each contributes to the final performance, demonstrating that they are critical for obtaining strong attacks that outperform state-of-the-art gradient-based methods.
>
> Finally, we would also like to highlight that we extend evolutionary attacks to two new objectives that, to the best of our knowledge, have not been explored before: (i) certified robustness and (ii) reducing conformal prediction set size. Both objectives are also independently relevant to other research communities.

---

> > ### Author Response · Authors · 2025-11-25
> >
> > **Q1: EvA performs on larger graphs**
> > We refer the reviewer to Figure 5, the rightmost figure, in which we compare the performance of EVA to PRBCD on the ogbn-arxiv dataset. We also provide additional results in Appendix C.3.
> >
> > We would like to again thank the reviewer for the time and thoughtful feedback. We believe our revisions and responses address the raised concerns, and we hope this will be reflected in the updated assessment. We are happy to respond to any additional questions or feedback.

---

### Official Review · Reviewer_tEfF · 2025-11-01

**Soundness:** 3
**Presentation:** 3
**Contribution:** 3
**Rating:** 4
**Confidence:** 4

**Summary:**

This paper introduces EvA, a black-box adversarial attack method for graph neural networks that uses a genetic algorithm to perturb the graph structure. Unlike gradient-based attacks, EvA directly optimizes discrete, non-differentiable objectives(such as classification accuracy) without relying on gradient approximations or domain relaxation. EvA highlights the limitations of gradient-based methods and establishes evolutionary search as a powerful, underexplored paradigm for adversarial attacks on graphs.

**Strengths:**

> 1. The paper's primary strength is its successful revival of evolutionary search, a paradigm previously dismissed as inferior. It demonstrates that with careful design, this approach can decisively outperform state-of-the-art gradient-based methods, challenging a core assumption in the field and opening a new direction for research.
> 2. The work is supported by comprehensive experiments and thorough ablation studies that validate every design choice.
> 3. The significance of the work is greatly amplified by applying EvA to novel objectives beyond accuracy, such as breaking robustness certificates and conformal predictions.

**Weaknesses:**

> 1. The paper rightly notes the high query complexity as a limitation, but does not conduct a rigorous quantitative trade-off analysis between computational cost and performance improvement. In my opinion, this is essential for the practical evaluation of the method.
> 2. The paper presents compelling empirical evidence for EvA's superiority over gradient-based methods. However, the explanatory depth for this success seems limited, primarily resting on the well-established notion of gradient unreliability. A more profound analysis examining whether EvA's advantage stems from superior navigation of non-convex loss landscapes or effective exploitation of higher-order edge interaction effects would significantly strengthen the work and provide foundational insights for future research. For more details, please refer to the question section.

**Questions:**

- While EVA demonstrates that Genetic Algorithms can achieve considerable effectiveness in conducting adversarial attacks, I still feel I don't fully grasp its fundamental mechanisms.

- In section 3, the sentence "We hypothesise that EvA, leveraging the exploratory capabilities of GA, can explore the search space more effectively and avoid local optima, while PRBCD gets stuck." "Exploratory capabilities" and "avoid local optima" are essentially standard claims for all GAs, bordering on being tautological. The paper seems to fail to specify how exactly the exploration capability of the EvA manifests in the specific context of discrete graph perturbation spaces. However, the analysis of perturbation patterns might provide the most relevant clues. As shown in Appendix D.1 "Label diversity", this section compares the statistical characteristics of perturbation solutions found by EvA and PRBCD. The analysis reveals that EvA's perturbation connections are more uniformly distributed across nodes with different labels, and demonstrate a greater tendency to connect to high-degree nodes and high-margin nodes. While this analysis is highly valuable, it primarily describes "what the solution looks like" rather than "how this solution was progressively discovered." It would be enlightening if the authors could demonstrate how the genetic algorithm guides the search process, which could potentially enhance the paper's readability and conceptual clarity.

---

> ### Author Response · Authors · 2025-11-25
>
> We thank the reviewer for taking the time to read our paper and for the thoughtful comments. We have outlined our responses below and hope they resolve your concerns.
>
> **W1: *trade-off analysis between computational cost and performance improvement.***
>
> We thank the reviewer for the thoughtful comment. To address the comparison in terms of the number of queries, we refer the reviewer to Figure 3. On the left, we show the comparison between the number of forward passes (queries) in EVA and the number of forward and backward passes in PRBCD. As the figure illustrates, for a smaller dataset such as CoraML, we can match PRBCD’s performance with fewer than 100 forward passes and subsequently outperform it. As the graph size increases, the problem becomes more challenging, which is intuitive. Consequently, for PubMed, we begin to outperform PRBCD after approximately 400 forward passes and continue to improve after that point.
>
> Additionally, we provide a comparison of memory and time complexity in Figure 10 (Appendix), which offers further ablation analysis.
>
> -----
>
> **W2, Q2: How this solution was progressively discovered**
>
> We agree with the reviewer, and to address this issue, we provide additional analysis in figure 10, 11, 12, 13 and 14 and table 19 and 20. First, as suggested by the reviewer, we track different steps of EvA and identify several interesting behaviours.  For example, In figure 10 we find that the EvA starts by connecting nodes with high-margin (confident) to those with low-margin (non-confident), which makes sense since low-margin nodes are easier to attack. As EvA progresses, it begins to connect more nodes with medium-margin to those with high-margin, which are more difficult to attack, but such perturbations could distort the representations of both nodes. We provide a more detailed analysis of the observed patterns over time, as well as a similar analysis w.r.t. the degree and label of the nodes in the newly section in the Appendix marked with green colour.
>
> We also conduct a further study on the solutions found by EvA and PRBCD over 10 different splits. We observe that in almost all runs, EvA directly influences more nodes compared to PRBCD (282 vs 208 nodes), by using less budget per node. In table 19 and 20, we observe that PRBCD usually spends more 2-flips and 3-flips compared to EvA and, in some cases, uses a higher budget (e.g., 10–30) on specific nodes, which means it overspends the budget in the hope of changing only that node’s label. We  provide the detail of each run in the Appendix.
>
> In figure 13, we also provide statistics on the union and the intersection of all edges and nodes across the entire population, at each step. This results shows how EvA repeatedly transitions between phases of exploration and exploitation. In addition, in figure 14 we show perturbation graphs over different steps, visualised in 2D via a tSNE embedding, which reveals several interesting patterns that we further explain in the paper.
>
> We invite the reviewer to consider the new observations, which we have highlighted in green in the updated manuscript.
>
> -----
>
> **Q1: EvA mechanisms:**
>
>  We updated our manuscript and added further pseudocode in appendix section D.1 to make our method clear for the reader.
>
> We would like to again thank the reviewer for the time and thoughtful feedback. We believe our revisions and responses address the raised concerns, and we hope this will be reflected in the updated assessment. We are happy to respond to any additional questions or feedback.

---

### Meta-Review · Area_Chair_g1Vq · 2026-01-08

**Summary:**

The article has two good reviews and two lacking reviews. Those lacking reviews are on the extreme ends with 2 and 8. The two other reviews are both rated 4 and one of them would likely increase to 6.

The article was criticized by the "2 reviewer" for its writing, and I can see some of their points. For example, table headers do not indicate which metric is being reported as "performance". An algorithm was added after the reviews, but the pseudocode uses variables that are neither input nor defined (such as delta). I am not sure if the reviewers would be satisfied with the additions.

However, the article revives the genetic search area in this domain, which is valuable.  The reviewers also note competitive results for the proposed model.

**Reviewer Concerns:**

tEfF
- Praises the work for its revival of the evolutionary search in adversarial attacks.
- Asks for cost analysis and criticizes the depth of explanations.  The authors respond by pointing to figs 3 and 10 that contain the asked analysis.
- The authors also prepare new figures to address the analysis question.

4Bnj
- Asks for a positioning of this attack wrt to prior heuristics and search-based attacks. The authors here miss the chance to prepare a new, well-crafted text to be inserted into the article.
- Asks for a better scalability and efficiency analysis, the authors respond with Figures 3 and 10 discussion.
- The reviewer is especially concerned about the heuristic nature of this work. This would be a major point for a rating consideration. The authors respond with existing discussion, but here an opportunity is missed again to present a new analysis to win over the reviewer.

oVZ5
- The reviewer asks for a formulation of a competitor method, and the authors rightfully respond that it is not standard practice to define previous work.
- The reviewer also criticizes the lack of prior work in comparisons, but fails to name any such work.
- In general, this review leaves much to be desired and does not pose any valid criticism.

kDQh
- praises the article for results and approach.
- criticizes the cost of the method. This review leaves much to be desired in constructive criticism as well.

**Reviewer Scores:**

tEfF rates 4 would increase to 6.
4Bnj rates 4, and would be sympathetic to an increase to 6, but given the response, it would be highly unlikely.
oVZ5 rated 2 and would not increase the rating.
kDQh rated 8 and would keep the score.

---

### Decision · Program_Chairs · 2026-01-26

Accept (Poster)